# SCALING FORWARD GRADIENT WITH LOCAL LOSSES

**Mengye Ren**[1*]**, Simon Kornblith**[2]**, Renjie Liao**[3]**, Geoffrey Hinton**[2,4]
[1]NYU, [2]Google, [3]UBC, [4]Vector Institute

## ABSTRACT

Forward gradient learning computes a noisy directional gradient and is a biologically plausible alternative to backprop for learning deep neural networks. However, the standard forward gradient algorithm, when applied naively, suffers from high variance when the number of parameters to be learned is large. In this paper, we propose a series of architectural and algorithmic modifications that together make forward gradient learning practical for standard deep learning benchmark tasks. We show that it is possible to substantially reduce the variance of the forward gradient estimator by applying perturbations to activations rather than weights. We further improve the scalability of forward gradient by introducing a large number of local greedy loss functions, each of which involves only a small number of learnable parameters, and a new MLPMixer-inspired architecture, LocalMixer, that is more suitable for local learning. Our approach matches backprop on MNIST and CIFAR-10 and significantly outperforms previously proposed backprop-free algorithms on ImageNet. Code is released at `https://github.com/google-research/google-research/tree/master/local_forward_gradient`.

## 1 INTRODUCTION

Most deep neural networks today are trained using the backpropagation algorithm (a.k.a. backprop) (Werbos, 1974; LeCun, 1985; Rumelhart et al., 1986), which efficiently computes the gradients of the weight parameters by propagating the error signal backwards from the loss function to each layer. Although artificial neural networks were originally inspired by biological neurons, backprop has always been considered as "biologically implausible" as the brain does not form symmetric backward connections or perform synchronized computations. From an engineering perspective, backprop is incompatible with a massive level of model parallelism, and restricts potential hardware designs. These concerns call for a drastically different learning algorithm for deep networks.

In the past, there have been attempts to address the above weight transport problem by introducing random backward weights (Lillicrap et al., 2016; Nøkland, 2016), but they have been found to scale poorly on larger datasets such as ImageNet (Bartunov et al., 2018). Addressing the issue of global synchronization, several papers showed that greedy local loss functions can be almost as good as end-to-end learning (Belilovsky et al., 2019; Löwe et al., 2019; Xiong et al., 2020). However, they still rely on backprop for learning a number of internal layers within each local module.

Approaches based on weight perturbation, on the other hand, directly send the loss signal back to the weight connections and hence do not require any backward weights. In the forward pass, the network adds a slight perturbation to the synaptic connections and the weight update is then multiplied by the negative change in the loss. Weight perturbation was previously proposed as a biologically plausible alternative to backprop (Xie & Seung, 1999; Seung, 2003; Fiete & Seung, 2006). Instead of directly perturbing the weights, it is also possible to use forward-mode automatic differentiation (AD) to compute a directional gradient of the final loss along the perturbation direction (Pearlmutter, 1994). Algorithms based on forward-mode AD have recently received renewed interest in the context of deep learning (Baydin et al., 2022; Silver et al., 2022). However, existing approaches suffer from the curse of dimensionality, and the variance of the estimated gradients is too high to effectively train large networks.

In this paper, we revisit activity perturbation (Le Cun et al., 1988; Widrow & Lehr, 1990; Fiete & Seung, 2006) as an alternative to weight perturbation. As previous works focused on specific settings, we explore the general applicability to large networks trained on challenging vision tasks.

---

*Work done as a visiting faculty researcher at Google. Correspondence to: `mengye@cs.nyu.edu`.

We prove that activity perturbation yields lower-variance gradient estimates than weight perturbation, and provide a continuous-time rate-based interpretation of our algorithm. We directly address the scalability issue of forward gradient learning by designing an architecture with many local greedy loss functions, isolating the network into local modules and hence reducing the number of learnable parameters per loss. Unlike prior work that only adds local losses along the depth dimension, we found that having patch-wise and channel group-wise losses is also critical. Lastly, inspired by the design of MLPMixer (Tolstikhin et al., 2021), we designed a network called LocalMixer, featuring a linear token mixing layer and grouped channels for better compatibility with local learning.

We evaluate our local greedy forward gradient algorithm on supervised and self-supervised image classification problems. On MNIST and CIFAR-10, our learning algorithm performs comparably with backprop, and on ImageNet, it performs significantly better than other biologically plausible alternatives using asymmetric forward and backward weights. Although we have not fully matched backprop on larger-scale problems, we believe that local loss design could be a critical ingredient for biologically plausible learning algorithms and the next generation of model-parallel computation.

## 2 RELATED WORK

Ever since the perceptron era, the design of learning algorithms for neural networks, especially algorithms that could be realized by biological brains, has been a central interest. Review papers by Whittington & Bogacz (2019) and Lillicrap et al. (2020) have systematically summarized the progress of biologically plausible deep learning. Here, we discuss related work in the following subtopics.

**Forward gradient and reinforcement learning.** Our work leverages forward-mode automatic differentiation (AD), which was first proposed by Wengert (1964). Later it was used to learn recurrent neural networks (Williams & Zipser, 1989) and to compute Hessian vector products (Pearlmutter, 1994). Computing the true gradient using forward-mode AD requires the full Jacobian, which is often large and expensive to compute. Recently, Baydin et al. (2022) and Silver et al. (2022) proposed to update the weights based on the directional gradient along a random or learned perturbation direction. They found that this approach is sufficient for small-scale problems. This general family of algorithms is also related to reinforcement learning (RL) and evolution strategies (ES), since in each case the network receives a global reward. RL and ES have a long history of application in neural networks (Whitley, 1993; Stanley & Miikkulainen, 2002; Salimans et al., 2017), and they are effective for certain continuous control and decision-making tasks. Clark et al. (2021) found global credit assignment can also work well in vector neural networks where weights are only present between vectorized groups of neurons.

**Greedy local learning.** There have been numerous attempts to use local greedy learning objectives for training deep neural networks. Greedy layerwise pretraining (Bengio et al., 2006; Hinton et al., 2006; Vincent et al., 2010) trains individual layers or modules one at a time to greedily optimize an objective. Local losses are typically applied to different layers or residual stages, using common supervised and self-supervised loss formulations (Belilovsky et al., 2019; Nøkland & Eidnes, 2019; Löwe et al., 2019; Belilovsky et al., 2020). Xiong et al. (2020); Gomez et al. (2020) proposed to use overlapped losses to reduce the impact of greedy learning. Patel et al. (2022) proposed to split a network into neuron groups. Laskin et al. (2020) applied greedy local learning on model parallelism training, and Wang et al. (2021) proposed to add a local reconstruction loss for preserving information. However, most local learning approaches proposed in the last decade rely on backprop to compute the weight updates within a local module. One exception is the work of Nøkland & Eidnes (2019), which avoided backprop by using layerwise objectives coupled with a similarity loss or a feedback alignment mechanism. Gated linear networks and their variants (Veness et al., 2017; 2021; Sezener et al., 2021) ask every neuron to make a prediction, and have shown interesting results on avoiding catastrophic forgetting. From a theoretical perspective, Baldi & Sadowski (2016) provided insights and proofs on why local learning can be worse than global learning.

**Asymmetric feedback weights.** Backprop relies on weight symmetry: the backward weights are the same as the forward weights. Past research has looked at whether this constraint is necessary. Lillicrap et al. (2016) proposed *feedback alignment* (FA) that uses random and fixed backward weights and found it can support error driven learning in neural networks. Direct FA (Nøkland, 2016) uses a single backward layer to wire the loss function back to each layer. There have also been methods that aim to explicitly update backward weights. Recirculation (Hinton & McClelland, 1987) and target propagation (TP) (Bengio, 2014; Lee et al., 2015; Bartunov et al., 2018) use local reconstruction

objective to learn separate forward and backward weights as approximate inverses of each other. Ladder networks (Rasmus et al., 2015) found local reconstruction objectives and asymmetric weights can help achieve strong semi-supervised learning performance. However, Bartunov et al. (2018) reported both FA and TP algorithms do not scale to larger problems such as ImageNet, where their error rates are over 90%. Liao et al. (2016); Xiao et al. (2019) proposed sign symmetry (SS) where each backward connection weight share the same sign as the forward counterpart. Akrout et al. (2019) proposed weight mirroring and the modified Kolen-Pollack algorithm (Kolen & Pollack, 1994) to align forward and backward weights. Woo et al. (2021) proposed to update using activities from several layers below to avoid bidirectional connections. Compared to these works, we circumvent the issue of weight symmetry, and more generally network symmetry, by using only reward (and the change rate thereof), instead of backward weights.

**Biologically plausible perturbation learning.** Forward gradient is related to perturbation learning in the biology context. Traditionally, neural plasticity learning rules focus on deriving weight updates as a function of the input and output activity of a neuron (Hebb, 1949; Widrow & Hoff, 1960; Oja, 1982; Bienenstock et al., 1982; Abbott & Nelson, 2000). Weight perturbation learning (Jabri & Flower, 1992), on the other hand, is much more general as it permits any form of global reward (Schultz et al., 1997). It was developed in both rated-based and spiking-based formuations (Xie & Seung, 1999; Seung, 2003). Activity (or node) perturbation was proposed in shallow networks (Le Cun et al., 1988; Widrow & Lehr, 1990) and later in a spike-based continuous time network (Fiete & Seung, 2006), where it was interpreted as the perturbation of the conductance of neurons. Werfel et al. (2003) showed that backprop has a faster convergence rate than perturbation learning, and activity perturbation wins over weight perturbation by another factor. In our work, we show activity perturbation has lower gradient estimation variance compared to weight perturbation.

## 3 FORWARD GRADIENT LEARNING

In this section, we review and establish the technical background for our learning algorithm. We first review the technique of forward-mode automatic differentiation (AD). Second, we formulate two different types of perturbation in the weight space or activity space.

### 3.1 FORWARD-MODE AUTOMATIC DIFFERENTIATION (AD)

Let $f : \mathbb{R}^m \mapsto \mathbb{R}^n$. The Jacobian of $f$, $J_f$, is a matrix of size $n \times m$. Forward-mode AD computes the matrix-vector product $J_f \mathbf{v}$, where $\mathbf{v} \in \mathbb{R}^m$. It is defined as the directional gradient along $\mathbf{v}$ evaluated at $\mathbf{x}$:

$$J_f \mathbf{v} := \lim_{\delta \mapsto 0} \frac{f(\mathbf{x} + \delta \mathbf{v}) - f(\mathbf{x})}{\delta}. \tag{1}$$

For comparison, backprop, also known as reverse-mode AD, computes the vector-Jacobian product $\mathbf{v} J_f$, where $\mathbf{v} \in \mathbb{R}^n$, which corresponds to the last term in the chain rule. In contrast to reverse-mode AD, forward-mode AD only requires one forward pass, which is augmented with the derivative information. To compute the Jacobian vector product of a node in a computation graph, first the input node will be augmented with $\mathbf{v}$, which is the vector to be multiplied. Then for other nodes, we send in a tuple of $(\mathbf{x}, \mathbf{x}')$ as inputs and compute a tuple $(\mathbf{y}, \mathbf{y}')$ as outputs, where $\mathbf{x}'$ and $\mathbf{y}'$ are the intermediate derivatives at node $\mathbf{x}$ and node $\mathbf{y}$, $i.e.$ $\mathbf{y}' = \frac{d\mathbf{y}}{d\mathbf{x}} \mathbf{x}'$, and $\frac{d\mathbf{y}}{d\mathbf{x}}$ is the Jacobian between $\mathbf{y}$ and $\mathbf{x}$. In the JAX library (Bradbury et al., 2018), forward-mode AD is implemented as `jax.jvp`.

### 3.2 WEIGHT-PERTURBED FORWARD GRADIENT

Weight perturbation to generate weight updates was originally explored in (Barto et al., 1983; Xie & Seung, 1999; Seung, 2003). Baydin et al. (2022) uses the technique of forward-mode AD to implement weight perturbation, which is better than finite differences in terms of numerical stability. Let $w_{ij}$ be the weight connection between unit $i$ and $j$, and $f$ be the loss function. We can estimate the gradient by sampling a random matrix with iid elements $v_{ij}$ drawn from a zero-mean unit-variance Gaussian distribution. The estimator is

$$g_w(w_{ij}) = \left( \sum_{i'j'} \nabla w_{i'j'} v_{i'j'} \right) v_{ij}. \tag{2}$$

Intuitively, this estimator samples a random perturbation direction $v_{ij}$ and tests how it aligns with the true gradient $\nabla w_{i'j'}$ by using forward-mode to perform the dot product, and then multiplies the scalar alignment with the perturbation direction again. Baydin et al. (2022) referred this form of gradient estimation using forward-mode AD as "forward gradient". To distinguish with another form

|  | Unbiased? | Avg. Variance (shared) | Avg. Variance (independent) |
|---|---|---|---|
| $g_w(\cdot)$ | Yes | $\frac{pq+2}{N}V + (pq+1)S$ | $\frac{pq+2}{N}V + \frac{pq+1}{N}S$ |
| $g_a(\cdot)$ | Yes | $\frac{q+2}{N}V + (q+1)S$ | $\frac{q+2}{N}V + \frac{q+1}{N}S$ |

Table 1: Comparing weight ($g_w$) and activity ($g_a$) perturbation. $V$=dimension-wise avg. gradient variance, $S$=dimension-wise avg. squared gradient norm; $p$=fan-in; $q$=fan-out; $N$=batch size.

of perturbation we detail later, we refer this to as "weight-perturbed forward gradient", or simply as "weight perturbation".

### 3.3 ACTIVITY-PERTURBED FORWARD GRADIENT

An alternative to perturbing the weights is to instead perturb the activities, which can reduce the number of perturbation dimensions per example. Activity perturbation was originally explored in Le Cun et al. (1988); Widrow & Lehr (1990) under restrictive assumptions. Here, we introduce a general way to estimate gradients using activity perturbation. It is potentially biologically plausible, since it could correspond to perturbation of the conductance in each neuron (Fiete & Seung, 2006). Here, we focus on a discrete-time rate-based formulation for simplicity. Let $x_i$ denote the activity of the $i$-th pre-synaptic neuron and $z_j$ denote that of the $j$-th post-synaptic neuron before the non-linear activation function, and $u_j$ be the perturbation of $z_j$. The activity-perturbed forward gradient estimator is

$$g_a(w_{ij}) = x_i \left( \sum_{j'} \nabla z_{j'} u_{j'} \right) u_j, \tag{3}$$

where the inner product between $\nabla \mathbf{z}$ and $\mathbf{u}$ is again computed by using forward-mode AD.

### 3.4 THEORETICAL PROPERTIES

In this section we aim to analyze the expectation and variance properties of forward gradient estimators. We focus our analysis on the gradient of one weight matrix $\{w_{ij}\}$, but the conclusion holds for a network with many weight matrices too.

Table 1 summarizes the theoretical results[1]. With a batch size of $N$, independent perturbation can achieve $1/N$ reduction of variance, whereas shared perturbation has a constant variance term dominated by the squared gradient norm. However, when performing independent weight perturbation, matrix multiplications cannot be batched because each example's activation vector is multiplied with a different weight matrix. By contrast, independent activity perturbation admits batched matrix multiplications. Moreover, activity perturbation enjoys a factor of fan-in ($p$) times smaller variance compared to weight perturbation since the number of perturbed elements is the number of output units instead of the size of the whole weight matrix. The only drawback of activity perturbation is the memory required for storage of intermediate activations, in exchange for a factor of $Np$ reduction in variance. However, for both activity and weight perturbation, the variance still grows with larger networks. In Section 4 we will further reduce the variance by introducing local loss functions.

### 3.5 CONTINUOUS-TIME RATE-BASED MODELS

Forward-mode AD can be viewed as computing the first-order time derivative in a continuous-time physical system. Suppose the tuples passed between nodes of the computation graph are $(\mathbf{x}, \dot{\mathbf{x}})$, where $\dot{\mathbf{x}}$ is the change in $\mathbf{x}$ over time. The computation is then the same as forward-mode AD. For each node, $\dot{\mathbf{y}} = \frac{d\mathbf{y}}{d\mathbf{x}}\dot{\mathbf{x}}$, where $\frac{d\mathbf{y}}{d\mathbf{x}}$ is the Jacobian between the output and the input. Note that in a physical system we don't have to explicitly perform the differentiation operation by running two forward passes. Instead the first-order derivative information is readily available in the analog signal, and we only need to plug the output signal into a differentiator circuit.

The activity-perturbed learning rule for a continuous time system is thus $\dot{w}_{ij} \propto x_i \dot{y}_j \dot{r}$, where $x_i$ is the pre-synaptic activity, and $\dot{y}_j$ is the rate of change in the post-synaptic activity, which is the perturbation direction for a small period of time, and $\dot{r}$ is the rate of change of reward (or the negative loss). The reward controls whether learning is Hebbian or anti-Hebbian. Both Hinton et al. (2007) and Bengio et al. (2017) propose to use a product of pre-synaptic activity and the rate of change of postsynaptic activity. However, they did not consider using the rate of change of reward as a modulator and instead relied on another set of feedback weights to communicate the error signal through inputs. In contrast, we show that by broadcasting the rate of change of reward, we can actually bypass the weight transport problem.

---

[1]All proofs can be found in Appendix 8 and 9. Numerical simulation results can be found in Appendix 10.

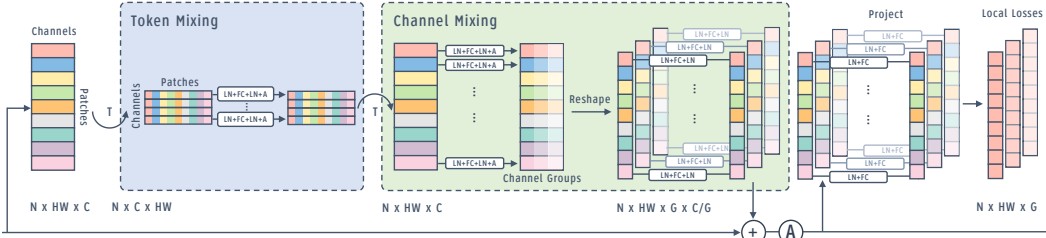

Figure 1: A LocalMixer network consists of several mixer blocks. A=Activation function (ReLU).

Figure 2: A LocalMixer residual block with local losses. Token mixing consists of a linear layer and channels are grouped in the channel mixing layers. Layer norm is applied before and after every linear layer. LN=Layer Norm; FC=Fully Connected layer; A=Activation function (ReLU); T=Transpose.

### 3.6 ACTIVATION SPARSITY AND NORMALIZATION FUNCTIONS

In networks with ReLU activations, we can leverage ReLU sparsity to achieve further variance reduction, because the inactivated units will have zero gradient and therefore we should not perturb these units, and set the perturbation to be zero.

Normalization layers are often added in deep neural networks after the linear layer. To compute the correct gradient in activity perturbation, we also need to account for normalization in the weight update rule. Since there is no backward weight connections, one option is to simply apply backprop on normalization layers. However, we also found that it is also fine to ignore the gradient of normalization layer when using layer normalization.

## 4 SCALING WITH LOCAL LOSSES

As we have explained in the previous section, perturbation learning can suffer from a curse of dimensionality: the variance grows with the number of perturbation dimensions, and in deep networks there are often millions of parameters changing at the same time. One way to limit the number of learnable dimensions is to divide the network into submodules, each with a separate loss function. In this section, we will explore several ways to increase the number of local losses to tame the variance.

**1) Blockwise loss.** First, we will divide the network into modules in depth. Each module consists of several layers. At the end of each module, we compute a loss function, and that loss is used to update the parameters in that module. This approach is equivalent of adding a "stop gradient" operator in between modules. Such local greedy losses were previously explored in Belilovsky et al. (2019) and Löwe et al. (2019).

**2) Patchwise loss.** Sensory input signals such as images have spatial dimensions. We will apply a separate loss patchwise along these spatial dimensions. In the Vision Transformer architecture (Vaswani et al., 2017; Dosovitskiy et al., 2021), each spatial token represents a patch in the image. In modern deep networks, parameters in each spatial location are often shared to improve data efficiency and reduce memory bandwidth utilization. Although naive weight sharing is not biologically plausible, we still consider shared weights in this work. It may be possible to mimic the effect of weight sharing by adding knowledge distillation (Hinton et al., 2015) losses in between patches.

**3) Groupwise loss.** Lastly, we turn to the channel dimension. To create multiple losses, we split the channels into a number of groups, and each group is attached to a loss function (Patel et al., 2022). To prevent groups from communicating between each other, channels are only connected to other channels within the same group. A grouped linear layer is computed as $z_{g,j} = \sum_i w_{g,i,j} x_{g,i}$, for individual group $g$. Whereas previous work used channel groups to improve computational efficiency (Krizhevsky et al., 2012; Ioannou et al., 2017; Xie et al., 2017), in our work, adding groups contributes to the total number of losses and thus reduces variance.

**Feature aggregators.** Naively applying losses separately to the spatial and channel dimensions leads to suboptimal performances, since each dimension contains only local information. For losses of

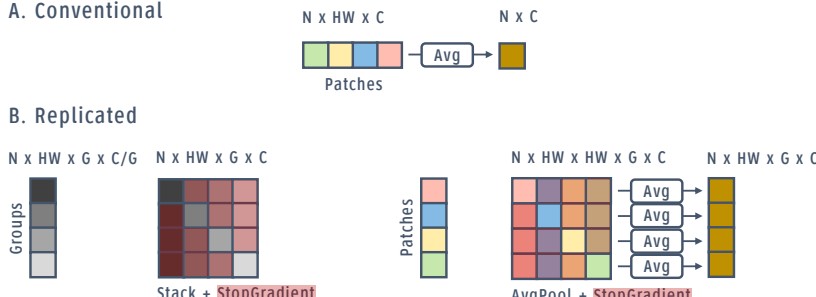

Figure 3: Feature aggregator designs. A) In the conventional design, average pooling is performed to aggregate features from different spatial locations. B) We propose the replicated design, features are first concatenated across groups and then averaged across spatial locations. We create copies of the same feature with different stop gradient masks so that we obtain more local losses instead of a global one. The stop gradient mask makes sure that perturbation in one spatial group corresponds to its loss function. The numerical value of the loss function is the same as the conventional design.

standard tasks such as classification, the model needs a global view of the inputs to make a decision. Standard architectures obtain this global view by performing global average pooling layer before the final classification layer. We therefore explore strategies for aggregating information from other groups and spatial patches before the local loss function.

We would prefer to perform aggregation without reducing the total number of dimensions. We thus propose a replicated design for feature aggregation, shown in Figure 3. First, channel groups are copied and communicated to one another, but every group except the active group itself is masked with stop gradient so that other groups do not affect the forward gradient computation:

$$\mathbf{x}_{p,g} = [\text{StopGrad}(x_{p,1}...x_{p,g-1}), x_{p,g}, \text{StopGrad}(x_{p,g+1}, ..., x_{p,G})], \tag{4}$$

where $p$ and $g$ index the patches and groups respectively. Similarly, each spatial location is also copied, communicated, and masked, and then averaged locally:

$$\overline{\mathbf{x}}_{p,g} = \frac{1}{P}\left(\mathbf{x}_{p,g} + \sum_{p' \neq p} \text{StopGrad}(\mathbf{x}_{p',g})\right). \tag{5}$$

The output of feature aggregation is the same as that of the conventional global average pooling layer. The difference is that here the loss is replicated and different patch groups are activated in each loss.

**Learning objectives.** We consider the supervised classification loss and the contrastive InfoNCE loss (van den Oord et al., 2018; Chen et al., 2020), which are the two most commonly used losses in image representation learning. For supervised classification, we attach a shared linear layer (shared across $p, g$) on top of the aggregated features for a cross entropy loss: $L_{p,g}^s = -\sum_k t_k \log \text{softmax}(W_l \overline{\mathbf{x}}_{p,g})_k$. The loss is of the same value across each group and patch location.

For contrastive learning, the linear layer becomes a linear feature projector. Suppose $\mathbf{x}_n^{(1)}$ and $\mathbf{x}_n^{(2)}$ are the two different views of the $n$-th example, the InfoNCE loss for contrastive learning is:

$$L_{p,g}^c = -\sum_n \log \frac{(W\overline{\mathbf{x}}_{n,p,g}^{(1)})^\top \text{StopGrad}(W\overline{\mathbf{x}}_n^{(2)})}{\sum_m (W\overline{\mathbf{x}}_{n,p,g}^{(1)})^\top \text{StopGrad}(W\overline{\mathbf{x}}_m^{(2)})}. \tag{6}$$

Note that we add a stop gradient operator on the second view. It is usually unnecessary to add this stop gradient in the InfoNCE loss; however, we found that perturbation-based methods require a stop gradient and otherwise the loss will not go down. This is likely because we share the perturbations on both views, and having the same perturbation will increase the dot product between the two views but is not desired from a representation learning perspective. Figure 4 shows a comparison of the loss curves. Non-shared perturbations also work but are worse than stop gradient.

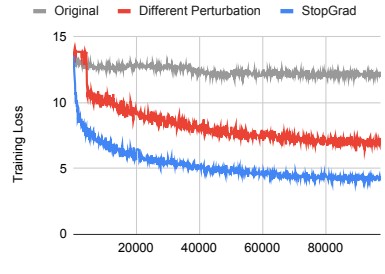

Figure 4: Importance of StopGradient in the InfoNCE loss, using M/8 on CIFAR-10 with 256 channels 1 group.

| Type | Blocks | Patches | Channels | Groups | Params | Dataset |
|---|---|---|---|---|---|---|
| LocalMixer S/1/1 | 1 | 1×1 | 256 | 1 | 272K | MNIST |
| LocalMixer M/1/16 | 1 | 1×1 | 512 | 16 | 429K | MNIST |
| LocalMixer M/8/16 | 4 | 8×8 | 512 | 16 | 919K | CIFAR-10 |
| LocalMixer L/8/64 | 4 | 8×8 | 2048 | 64 | 13.1M | CIFAR-10 |
| LocalMixer L/32/64 | 4 | 32×32 | 2048 | 64 | 17.3M | ImageNet |

Table 2: LocalMixer Architecture Details

## 5 IMPLEMENTATION

**Network architecture.** We propose the LocalMixer architecture that is more suitable for local learning. It takes inspiration from MLPMixer (Tolstikhin et al., 2021), which consists of fully connected networks and residual blocks. We leverage the fully connected networks so that each spatial patch performs computations without interfering with other patches, which is more compatible with our local learning objective. An image is divided into non-overlapping patches (*i.e.* tokens), and each block consists of token and channel mixing layers. Figure 1 shows the high level architecture, and Figure 2 shows the detailed diagram for one residual block. We add a linear projector/classification layer to attach a loss function at the end of each block. The last layer always uses backprop to update weights. For token mixing layers, we use one linear fully connected layer instead of an MLP, since we would like to make each block as shallow as possible. Before the last channel mixing layer, features are reshaped into a number of groups, and the last layer is fully connected within each feature group. Table 2 shows architectural details for the different sizes of models we investigate.

**Normalization.** There are many ways of performing normalization within a neural network across different tensor dimensions (Krizhevsky et al., 2012; Ioffe & Szegedy, 2015; Ba et al., 2016; Ren et al., 2017; Wu & He, 2018). We opted for a local variant of layer normalization that normalizes within each local spatial patch of features (Ren et al., 2017). For grouped linear layers, each group is normalized separately (Wu & He, 2018). Empirically, we found such local normalization performs better on contrastive learning experiments and about the same as layer normalization on supervised experiments. Local normalization is also more biologically plausible as it does not perform global communication. Conventionally, normalization layers are placed after linear layers. In MLPMixer (Tolstikhin et al., 2021), layer normalization is placed at the beginning of each residual block. We found it is the best to place normalization *before* and *after* each linear layer, as shown in Figure 2. Empirically this design choice does not make much difference for backprop, but it allows forward gradient learning to learn much faster and achieve lower training errors.

**Efficient implementation of replicated losses.** Due to the design of feature aggregation and replicated losses, a naïve implementation of groups can be very inefficient in terms of both memory consumption and compute. However, each spatial group actually computes the same aggregated feature and loss function. This means that it is possible to share most of the computation across loss functions when performing both backprop and forward gradient. We implemented our custom JAX JVP/VJP functions (Bradbury et al., 2018) and observed significant memory savings and compute speed-ups for replicated losses, which would otherwise not be feasible to run on modern hardware. The results are reported in Figure 5. A code snippet is included in Appendix 12.

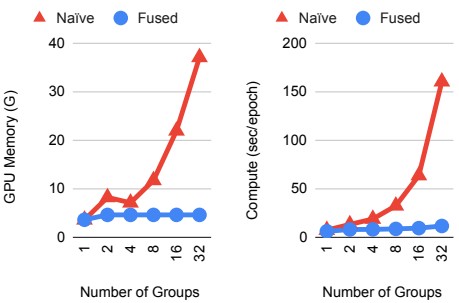

Figure 5: Memory and compute usage of naïve and fused implementation of replicated losses.

## 6 EXPERIMENTS

We compare our proposed algorithm to a set of alternatives: Backprop, Feedback Alignment and other global variants of Forward Gradient. Backprop is a biologically implausible oracle, since it computes true gradients whereas we compute noisy gradients. Feedback alignment computes approximate gradients by using a set of random backward weights. We explain each method below.

**1) Backprop (BP).** We include the standard backprop algorithm as well as its local variants. **Local Backprop (L-BP)** adds local losses as proposed, but still permits gradient to flow in an end-to-end fashion. **Local Greedy Backprop (LG-BP)** in addition adds stop gradient operators in between blocks. This is to provide a comparison to our methods by computing true local gradients. LG-BP is similar in spirit to recent local learning algorithms (Belilovsky et al., 2019; Löwe et al., 2019).

| Dataset
Network
Metric | MNIST
S/1/1
Test / Train Err. (%) | MNIST
M/1/16
Test / Train Err. (%) | CIFAR-10
M/8/16
Test / Train Err. (%) | ImageNet
L/32/64
Test / Train Err. (%) |
|---|---|---|---|---|
| BP | 2.66 / 0.00 | 2.41 / 0.00 | 33.62 / 0.00 | 36.82 / 14.69 |
| L-BP | 2.38 / 0.00 | 2.16 / 0.00 | 30.75 / 0.00 | 42.38 / 22.80 |
| LG-BP | 2.43 / 0.00 | 2.81 / 0.00 | 33.84 / 0.05 | 54.37 / 39.66 |
| BP-free algorithms | | | | |
| FA | **2.82** / **0.00** | 2.90 / **0.00** | 39.94 / 28.44 | 94.55 / 94.13 |
| L-FA | 3.21 / **0.00** | 2.90 / **0.00** | 39.74 / 28.98 | 87.20 / 85.69 |
| LG-FA | 3.11 / **0.00** | **2.50** / **0.00** | 39.73 / 32.32 | 85.45 / 82.83 |
| DFA | 3.31 / **0.00** | 3.17 / **0.00** | 38.80 / 33.69 | 91.17 / 90.28 |
| FG-W | 9.25 / 8.93 | 8.56 / 8.64 | 55.95 / 54.28 | 97.71 / 97.58 |
| FG-A | 3.24 / 1.53 | 3.76 / 1.75 | 59.72 / 41.29 | 98.83 / 98.80 |
| LG-FG-W | 9.25 / 8.93 | 5.66 / 4.59 | 52.70 / 51.71 | 97.39 / 97.29 |
| LG-FG-A | 3.24 / 1.53 | 2.55 / **0.00** | **30.68** / **19.39** | **58.37** / **44.86** |

Table 3: Supervised learning for image classification

| Dataset
Network
Metric | CIFAR-10
M/8/16
Test / Train Err. (%) | CIFAR-10
L/8/64
Test / Train Err. (%) | ImageNet
L/32/64
Test / Train Err. (%) |
|---|---|---|---|
| BP | 24.11 / 21.08 | 17.53 / 13.35 | 55.66 / 49.79 |
| L-BP | 24.69 / 21.80 | 19.13 / 13.60 | 59.11 / 52.50 |
| LG-BP | 29.63 / 25.60 | 23.62 / 16.80 | 68.36 / 62.53 |
| BP-free algorithms | | | |
| FA | 45.87 / 44.06 | 67.93 / 65.32 | 82.86 / 80.21 |
| L-FA | 37.73 / 36.13 | 31.05 / 26.97 | 83.18 / 79.80 |
| LG-FA | 36.72 / 34.06 | 30.49 / 25.56 | 82.57 / 79.53 |
| DFA | 46.09 / 42.76 | 39.26 / 37.17 | 93.51 / 92.51 |
| FG-W | 53.37 / 51.56 | 50.45 / 45.64 | 91.94 / 89.69 |
| FG-A | 54.59 / 52.96 | 56.63 / 56.09 | 97.83 / 97.79 |
| LG-FG-W | 52.66 / 50.23 | 52.27 / 48.67 | 91.36 / 88.81 |
| LG-FG-A | **32.88 / 29.73** | **26.81 / 23.90** | **73.24 / 66.89** |

Table 4: Self-supervised contrastive learning with linear readout

**2) Feedback Alignment (FA).** The standard FA algorithm (Lillicrap et al., 2016) adds a set of random and fixed backward weights. We assume that the gradients to normalization layers and activation functions are known since they do not have weight connections. Local Feedback Alignment (**L-FA**) adds local losses as proposed, but still permits error signals to flow back. Local Greedy Feedback Alignment (**LG-FA**) adds a stop gradient to prevent error signals from flowing back, similar to the backprop-free algorithm in Nøkland & Eidnes (2019).

**3) Forward Gradient (FG).** This family of methods comprises our proposed algorithm and related approaches. Weight-perturbed forward gradient (**FG-W**) was proposed by Baydin et al. (2022). In this paper, we propose the activity perturbation variant (**FG-A**). We further add local objective functions, producing **LG-FG-W** and **LG-FG-A**, which stand for Local Greedy Forward Gradient Weight/Activity-Perturbed. For local perturbation to work, we have to add a stop gradient in between blocks so each perturbation has a single corresponding loss. We expect **LG-FG-A** to achieve the best performance among other variants because it can leverage the variance reduction benefit from both activity perturbation and local losses.

**Datasets.** We use standard image classification datasets to benchmark the learning algorithms. MNIST (LeCun, 1998) contains 70,000 $28 \times 28$ handwritten digit images of class 0-9. CIFAR-10 (Krizhevsky et al., 2009) contains 60,000 $32 \times 32$ natural images of 10 semantic classes. ImageNet (Deng et al., 2009) contains 1.3 million natural images of 1000 classes, which we resized to $224 \times 224$. For CIFAR-10 and ImageNet, we applied both supervised learning and contrastive learning. For MNIST, we applied supervised learning only. We designed different configurations of the LocalMixer architecture for each dataset, listed in Table 2.

**Data augmentation.** For MNIST and CIFAR-10 supervised experiments, we do not apply data augmentation. Data augmentation on ImageNet follows the open source implementation by Grill

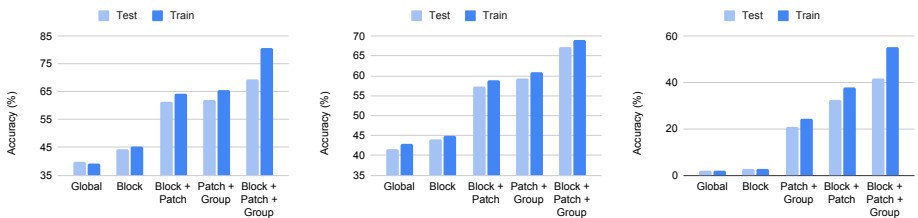

(a) CIFAR-10 Supervised M/8   (b) CIFAR-10 Contrastive M/8   (c) ImageNet Supervised L/32

Figure 6: Effect of adding local losses at different locations on the performance of forward gradient

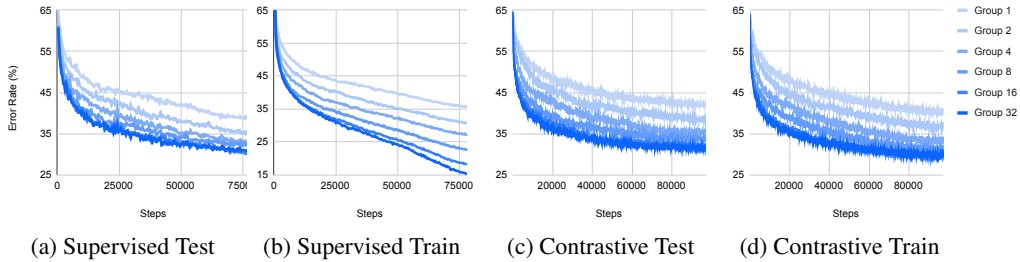

(a) Supervised Test   (b) Supervised Train   (c) Contrastive Test   (d) Contrastive Train

Figure 7: Error rate of M/8/* during CIFAR-10 training using different number of groups.

et al. (2020). Because forward gradient suffers from variance, we apply weaker augmentations for contrastive learning experiments, increasing the area lower bound for random crops from 0.08 to 0.3-0.5. We find that this change has relatively little effect on the performance of backprop.

**Main results.** Our main results are shown in Table 3 and Table 4. In supervised experiments, there is almost no cost of introducing local greedy losses, and our local forward gradient method can match the test error of backprop on MNIST and CIFAR. Note that LG-FG-A fails to overfit the training set to 0% error when trained without data augmentation. This suggests that variance could still be an issue. For CIFAR-10 contrastive learning, our method obtains an error rate approaching that obtained by backprop (26.81% vs. 17.53%), and most of the gap is due to greedy learning vs. gradient estimation (6.09% vs. 3.19%). On ImageNet, we achieve reasonable performance compared to backprop (58.37% vs. 36.82% for supervised and 73.24% vs. 55.66% for contrastive). However, we find that the error due to greediness grows as the problem gets more complex and requires more layers to cooperate. We significantly outperform the FA family on ImageNet (by 25% for supervised and 10% for contrastive). Interestingly, local greedy FA also performs better than global feedback alignment, which suggests that the benefit of local learning transfers to other types of gradient approximation. TP-based methods were evaluated in Bartunov et al. (2018) and were found to be worse than FA on ImageNet. In sum, although there is still some noticeable gap between our method and backprop, we have made a large stride forward compared to backprop-free algorithms. More results are included in the Appendix 14.

**Effect of local losses.** In Figure 6 we ablate the benefit of placing local losses at different locations: blockwise, patchwise and groupwise. A combination of all three is the strongest. Global perturbation learning fails to learn as the accuracy is similar to initializing with random weights.

**Effect of groups.** In Figure 7 we investigate the effect of different number of groups by showing the training curves. Adding more groups bring significant improvement to local perturbation learning in terms of lowering both training and test errors, but the effect vanishes around 8 channels / group.

## 7  CONCLUSION

It is often believed that perturbation-based learning cannot scale to large and deep networks. We show that this is to some extent true because the gradient estimation variance grows with the number of hidden dimensions for activity perturbation, and is even worse for shared weight perturbation. But more optimistically, we show that a huge number of local greedy losses can help forward gradient learning scale much better. We explored blockwise, patchwise, and groupwise local losses, and a combination of all three, with a total of a quarter of a million losses in one of the larger networks, performs the best. Local activity-perturbed forward gradient performs better than previous backprop-free algorithms on larger networks. The idea of local losses opens up opportunities for different loss designs and sheds light on the search for biologically plausible learning algorithms in the brain and alternative computing devices.

ACKNOWLEDGMENT

We thank Timothy Lillicrap for his helpful feedback on our earlier draft.

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

## 8 PROOFS OF UNBIASEDNESS

In this section, we show the unbiasedness of $g_w(w_{ij})$ and $g_a(w_{ij})$. The first proof was given by Baydin et al. (2022).

**Proposition 1.** $g_w(w_{ij})$ *is an unbiased gradient estimator if* $\{v_{ij}\}$ *are independent zero-mean uni-variance random variables (Baydin et al., 2022).*

*Proof.* We can rewrite the weight perturbation estimator as

$$g_w(w_{ij}) = \left( \sum_{i'j'} \nabla w_{i'j'} v_{i'j'} \right) v_{ij} = \nabla w_{ij} v_{ij}^2 + \sum_{i'j' \neq ij} \nabla w_{i'j'} v_{ij} v_{i'j'}. \tag{7}$$

Note that since each dimension of $v$ is an independent zero-mean uni-variance random variable, $\mathbb{E}[v_{ij}] = 0$, $\mathbb{E}[v_{ij}^2] = \text{Var}[v_{ij}] + \mathbb{E}[v_{ij}]^2 = 1 + 0 = 1$, and $\mathbb{E}[v_{ij}v_{i'j'}] = 0$ if $ij \neq i'j'$.

$$\mathbb{E}[g_w(w_{ij})] = \mathbb{E}\left[ \nabla w_{ij} v_{ij}^2 \right] + \mathbb{E}\left[ \sum_{i'j' \neq ij} \nabla w_{i'j'} v_{ij} v_{i'j'} \right] \tag{8}$$

$$= \nabla w_{ij} \, \mathbb{E}\left[ v_{ij}^2 \right] + \sum_{i'j' \neq ij} \nabla w_{i'j'} \, \mathbb{E}\left[ v_{ij} v_{i'j'} \right] \tag{9}$$

$$= \nabla w_{ij} \cdot 1 + \sum_{i'j' \neq ij} \nabla w_{i'j'} \cdot 0 \tag{10}$$

$$= \nabla w_{ij}. \tag{11}$$

$\square$

**Proposition 2.** $g_a(w_{ij})$ *is an unbiased gradient estimator if* $\{u_j\}$ *are independent zero-mean uni-variance random variables.*

*Proof.* The true gradient to the weights $\nabla w_{ij}$ is the product between $x_j$ and $\nabla z_k$. Therefore, we can rewrite the weight perturbation estimator as

$$g_a(w_{ij}) = x_i \left( \sum_{j'} \nabla z_{j'} u_{j'} \right) u_j = x_j \nabla z_j u_j^2 + x_i \left( \sum_{j' \neq j} \nabla z_{j'} u_{j'} \right) u_j \tag{12}$$

$$= x_i \nabla z_j u_j^2 + \left( \sum_{j' \neq j} x_i \nabla z_{j'} u_{j'} \right) u_j \tag{13}$$

$$= \nabla w_{ij} u_j^2 + \left( \sum_{j' \neq j} \nabla w_{ij'} u_{j'} \right) u_j. \tag{14}$$

Since each dimension of $u$ is an independent zero-mean uni-variance random variable, $\mathbb{E}[u_j] = 0$, $\mathbb{E}[u_j^2] = \text{Var}[u_j] + \mathbb{E}[u_j]^2 = 1 + 0 = 1$, and $\mathbb{E}[u_j u_{j'}] = 0$ if $j \neq j'$.

$$\mathbb{E}[g_a(w_{ij})] = \mathbb{E}\left[ \nabla w_{ij} u_j^2 + \left( \sum_{j' \neq j} \nabla w_{ij'} u_{j'} \right) u_j \right] \tag{15}$$

$$= \nabla w_{ij} \, \mathbb{E}\left[ u_j^2 \right] + \sum_{j' \neq j} \nabla w_{ij'} \, \mathbb{E}\left[ u_j u_{j'} \right] \tag{16}$$

$$= \nabla w_{ij} \cdot 1 + \sum_{j' \neq j} \nabla w_{ij'} \cdot 0 \tag{17}$$

$$= \nabla w_{ij}. \tag{18}$$

$\square$

## 9 PROOFS OF VARIANCES

We followed Wen et al. (2018) and show that the variance of the gradient estimators can be decomposed.

**Lemma 1.** *The variance of the gradient estimator can be decomposed into three parts:* $\mathrm{Var}\left(g(w_{ij})|x\right) = Z_1 + Z_2 + Z_3$, *where* $Z_1 = \frac{1}{N} V_1 \mathrm{Var}_x\left(\nabla w_{ij}|_x\right)$, $Z_2 = \frac{1}{N} \mathbb{E}_x\left[\mathrm{Var}_v\left(g(w_{ij})|x\right)\right]$, $Z_3 = \frac{1}{N^2} \mathbb{E}_B\left[\sum_{x^{(n)} \in B} \sum_{x^{(m)} \in B \setminus \{x^{(n)}\}} \mathrm{Cov}_v(g(w_{ij})|x^{(n)}, g(w_{ij})|x^{(m)})\right]$.

*Proof.* By the law of total variance,

$$\mathrm{Var}\left(g(w_{ij})\right) = \mathrm{Var}_B\left(\mathbb{E}_v\left[g(w_{ij})|B\right]\right) + \mathbb{E}_B\left[\mathrm{Var}_v(g(w_{ij})|B)\right]. \tag{19}$$

The first term comes from the gradient variance from data sampling, and it vanishes as batch size grows:

$$\mathrm{Var}_B\left(\mathbb{E}_v\left[g(w_{ij})|B\right]\right) \tag{20}$$

$$= \mathrm{Var}_B\left(\mathbb{E}_v\left[\frac{1}{N}\sum_{x^{(n)} \in B} g(w_{ij})|x^{(n)}\right]\right) \tag{21}$$

$$= \frac{1}{N^2}\mathrm{Var}_B\left(\mathbb{E}_v\left[\sum_{x^{(n)} \in B} g(w_{ij})|x^{(n)}\right]\right) \tag{22}$$

$$= \frac{1}{N^2}\mathrm{Var}_B\left(\sum_{x^{(n)} \in B} \mathbb{E}_v\left[g(w_{ij})|x^{(n)}\right]\right) \tag{23}$$

$$= \frac{1}{N^2}\mathrm{Var}_B\left(\sum_{x^{(n)} \in B} \nabla w_{ij}|_{x^{(n)}}\right) \tag{24}$$

$$= \frac{1}{N^2}\sum_n \mathrm{Var}_x\left(\nabla w_{ij}|_x\right) = \frac{1}{N}\mathrm{Var}_x\left(\nabla w_{ij}|_x\right) = Z_1. \tag{25}$$

The second term comes from the gradient estimation variance:

$$\mathbb{E}_B\left[\mathrm{Var}_v\left(g(w_{ij})|B\right)\right] \tag{26}$$

$$= \mathbb{E}_B\left[\mathrm{Var}_v\left(\frac{1}{N}\sum_{x^{(n)} \in B} g(w_{ij})\,\bigg|\,x^{(n)}\right)\right] \tag{27}$$

$$= \mathbb{E}_B\left[\frac{1}{N^2}\mathrm{Var}_v\left(\sum_{x^{(n)} \in B} g(w_{ij})\,\bigg|\,x^{(n)}\right)\right] \tag{28}$$

$$= \mathbb{E}_B\left[\frac{1}{N^2}\sum_{x^{(n)} \in B} \mathrm{Var}_v\left(g(w_{ij})|x^{(n)}\right) + \sum_{x^{(n)} \in B}\sum_{x^{(m)} \in B \setminus \{x^{(n)}\}} \mathrm{Cov}_v(g(w_{ij})|x^{(n)}, g(w_{ij})|x^{(m)})\right] \tag{29}$$

$$= \frac{1}{N}\mathbb{E}_x\left[\mathrm{Var}_v\left(g(w_{ij})|x\right)\right] + \frac{1}{N^2}\mathbb{E}_B\left[\sum_{x^{(n)} \in B}\sum_{x^{(m)} \in B \setminus \{x^{(n)}\}} \mathrm{Cov}_v(g(w_{ij})|x^{(n)}, g(w_{ij})|x^{(m)})\right] \tag{30}$$

$$= Z_2 + Z_3. \tag{31}$$

$\square$

**Remark.** *$Z_2$ is the variance of the gradient estimator in the deterministic case, and $Z_3$ measures the correlation between different gradient estimation within the batch. The $Z_3$ is zero if the perturbations are independent, and non-zero if the perturbations are shared within the mini-batch.*

**Proposition 3.** *Let $p \times q$ be the size of the weight matrix, the element-wise average variance of the weight perturbed gradient estimator with a batch size $N$ is $\frac{pq+2}{N}V + (pq+1)S$ if the perturbations are shared across the batch, and $\frac{pq+2}{N}V + \frac{pq+1}{N}S$ if they are independent, where $V$ is the element-wise average variance of the true gradient, and $S$ is the element-wise average squared gradient.*

*Proof.* We first derive $Z_2$.

$$Z_2 = \frac{1}{N}\underset{x}{\mathbb{E}}\left[\underset{v}{\mathrm{Var}}\left(g_w(w_{ij})\,\middle|\,x\right)\right] \tag{32}$$

$$= \frac{1}{N}\underset{x}{\mathbb{E}}\left[\underset{v}{\mathrm{Var}}\left(\left(\sum_{i'j'}\nabla w_{i'j'}v_{i'j'}\right)v_{ij}\right)\right] \tag{33}$$

$$= \frac{1}{N}\underset{x}{\mathbb{E}}\left[\underset{v}{\mathrm{Var}}\left(\nabla w_{ij}v_{ij}^2 + \sum_{i'j'\neq ij}\nabla w_{i'j'}v_{ij}v_{i'j'}\right)\right] \tag{34}$$

$$= \frac{1}{N}\underset{x}{\mathbb{E}}\left[\underset{v}{\mathrm{Var}}\left(\nabla w_{ij}v_{ij}^2\right) + \underset{v}{\mathrm{Var}}\left(\sum_{i'j'\neq ij}\nabla w_{i'j'}v_{ij}v_{i'j'}\right) + 2\underset{v}{\mathrm{Cov}}\left(\nabla w_{ij}v_{ij}^2, \sum_{i'j'\neq ij}\nabla w_{ij}v_{ij}v_{i'j'}\right)\right] \tag{35}$$

$$= \frac{1}{N}\underset{x}{\mathbb{E}}\left[\mathrm{Var}\left(\nabla w_{ij}v_{ij}^2\right) + \underset{v}{\mathrm{Var}}\left(\sum_{i'j'\neq ij}\nabla w_{i'j'}v_{ij}v_{i'j'}\right) + \right. \tag{36}$$

$$\left. 2\underset{v}{\mathbb{E}}\left[\sum_{i'j'\neq ij}\nabla w_{ij}\nabla w_{i'j'}v_{ij}^3 v_{i'j'}\right] - 2\underset{v}{\mathbb{E}}\left[\nabla w_{ij}v_{ij}^2\right]\underset{v}{\mathbb{E}}\left[\sum_{i'j'\neq ij}\nabla w_{i'j'}v_{ij}v_{i'j'}\right]\right] \tag{37}$$

$$= \frac{1}{N}\underset{x}{\mathbb{E}}\left[\nabla w_{ij}^2\underset{v}{\mathrm{Var}}\left(v_{ij}^2\right) + \underset{v}{\mathrm{Var}}\left(\sum_{i'j'\neq ij}\nabla w_{i'j'}v_{ij}v_{i'j'}\right) + \right. \tag{38}$$

$$\left. 2\sum_{i'j'\neq ij}\nabla w_{ij}\nabla w_{i'j'}\underset{v}{\mathbb{E}}\left[v_{ij}^3 v_{i'j'}\right] - 2\nabla w_{ij}\underset{v}{\mathbb{E}}\left[v_{ij}^2\right]\left(\sum_{i'j'\neq ij}\nabla w_{i'j'}\underset{v}{\mathbb{E}}\left[v_{ij}v_{i'j'}\right]\right)\right] \tag{39}$$

$$= \frac{1}{N}\underset{x}{\mathbb{E}}\left[\nabla w_{ij}^2\underset{v}{\mathrm{Var}}\left(v_{ij}^2\right) + \underset{v}{\mathrm{Var}}\left(\sum_{i'j'\neq ij}\nabla w_{i'j'}v_{ij}v_{i'j'}\right) + \right. \tag{40}$$

$$\left. 2\sum_{i'j'\neq ij}\nabla w_{ij}\nabla w_{i'j'}\cdot 0 - 2\nabla w_{ij}\cdot 1\left(\sum_{i'j'\neq ij}\nabla w_{i'j'}\cdot 0\right)\right] \tag{41}$$

$$= \frac{1}{N}\underset{x}{\mathbb{E}}\left[\nabla w_{ij}^2\underset{v}{\mathrm{Var}}\left(v_{ij}^2\right) + \underset{v}{\mathrm{Var}}\left(\sum_{i'j'\neq ij}\nabla w_{i'j'}v_{ij}v_{i'j'}\right)\right] \tag{42}$$

$$= \frac{1}{N}\underset{x}{\mathbb{E}}\left[\nabla w_{ij}^2\cdot(\mathbb{E}[v_{ij}^4] - \underset{v}{\mathbb{E}}[v_{ij}^2]^2) + \sum_{i'j'\neq ij}\underset{v}{\mathrm{Var}}\left(\nabla w_{i'j'}v_{ij}v_{i'j'}\right)\right] \tag{43}$$

$$= \frac{1}{N}\underset{x}{\mathbb{E}}\left[\nabla w_{ij}^2(3\underset{v}{\mathrm{Var}}[v_{ij}]^2 - \underset{v}{\mathbb{E}}[v_{ij}^2]^2) + \sum_{i'j'\neq ij}\nabla w_{i'j'}^2\underset{v}{\mathrm{Var}}\left(v_{ij}v_{i'j'}\right)\right] \tag{44}$$

$$= \frac{1}{N}\underset{x}{\mathbb{E}}\left[2\nabla w_{ij}^2\right] + \frac{1}{N}\underset{x}{\mathbb{E}}\left[\sum_{i'j'\neq ij}\nabla w_{i'j'}^2(\underset{v}{\mathrm{Var}}[v_{ij}] + \underset{v}{\mathbb{E}}[v_{i'j'}]^2)(\underset{v}{\mathrm{Var}}[v_{i'j'}] + \underset{v}{\mathbb{E}}[v_{i'j'}]^2) - \underset{v}{\mathbb{E}}[v_{ij}]^2\underset{v}{\mathbb{E}}[v_{i'j'}]^2\right] \tag{45}$$

$$= \frac{2}{N} \overline{\nabla w}_{ij}^2 + \frac{2}{N} \operatorname*{Var}_x(\nabla w_{ij}|x) + \frac{1}{N} \operatorname*{\mathbb{E}}_x \left[ \sum_{i'j' \neq ij} \nabla w_{i'j'}^2 \operatorname*{Var}_v(v_{ij}) \operatorname*{Var}_v(v_{i'j'}) \right] \tag{46}$$

$$= \frac{2}{N} \overline{\nabla w}_{ij}^2 + \frac{2}{N} \operatorname*{Var}_x(\nabla w_{ij}|x) + \frac{1}{N} \sum_{i'j' \neq ij} \operatorname*{\mathbb{E}}_x \left[ \nabla w_{i'j'}^2 \right] \tag{47}$$

$$= \frac{1}{N} \left[ \overline{\nabla w}_{ij}^2 + \operatorname*{Var}_x(\nabla w_{ij}|x) + \sum_{i'j'} \left( \overline{\nabla w}_{i'j'}^2 + \operatorname*{Var}_x(\nabla w_{i'j'}|x) \right) \right]. \tag{48}$$

$Z_3$ is nonzero if the perturbations are shared within a batch. Assuming that the perturbations are shared,

$$Z_3 = \frac{1}{N^2} \operatorname*{\mathbb{E}}_B \left[ \sum_{x^{(n)} \in B} \sum_{x^{(m)} \in B \setminus \{x^{(n)}\}} \operatorname*{Cov}_v( g_w(w_{ij})|\, x^{(n)}, g_w(w_{ij})|\, x^{(m)}) \right] \tag{49}$$

$$= \frac{1}{N^2} \operatorname*{\mathbb{E}}_B \left[ \sum_{x^{(n)} \in B} \sum_{x^{(m)} \in B \setminus \{x^{(n)}\}} \operatorname*{\mathbb{E}}_v \left[ g_w(w_{ij})|\, x^{(n)} \, g_w(w_{ij})|\, x^{(m)} \right] - \operatorname*{\mathbb{E}}_v \left[ g_w(w_{ij})|\, x^{(n)} \right] \operatorname*{\mathbb{E}}_v \left[ g_w(w_{ij})|\, x^{(m)} \right] \right] \tag{50}$$

$$= \frac{1}{N^2} \operatorname*{\mathbb{E}}_B \left[ \sum_{x^{(n)} \in B} \sum_{x^{(m)} \in B \setminus \{x^{(n)}\}} \operatorname*{\mathbb{E}}_v \left[ g_w(w_{ij})|\, x^{(n)} \, g_w(w_{ij})|\, x^{(m)} \right] - \nabla w_{ij}|x^{(n)} \nabla w_{ij}|x^{(m)} \right] \tag{51}$$

$$= \frac{1}{N^2} \operatorname*{\mathbb{E}}_B \left[ \sum_{x^{(n)} \in B} \sum_{x^{(m)} \in B \setminus \{x^{(n)}\}} \operatorname*{\mathbb{E}}_v \left[ \left( \sum_{i'j'} \nabla w_{i'j'}|x^{(n)} v_{i'j'} \right) v_{ij} \left( \sum_{i'j'} \nabla w_{i'j'}|x^{(m)} v_{i'j'} \right) v_{ij} \right] - \right. \tag{52}$$

$$\left. \nabla w_{ij}|x^{(n)} \nabla w_{ij}|x^{(m)} \right] \tag{53}$$

$$= \frac{1}{N^2} \operatorname*{\mathbb{E}}_B \left[ \sum_{x^{(n)} \in B} \sum_{x^{(m)} \in B \setminus \{x^{(n)}\}} \operatorname*{\mathbb{E}}_v \left[ \left( \nabla w_{ij}|x^{(n)} v_{ij}^2 + \sum_{i'j' \neq ij} \nabla w_{i'j'}|x^{(n)} v_{i'j'} v_{ij} \right) \right. \right. \tag{54}$$

$$\left. \left. \left( \nabla w_{ij}|x^{(m)} v_{ij}^2 + \sum_{i'j' \neq ij} \nabla w_{i'j'}|x^{(m)} v_{i'j'} v_{ij} \right) \right] \right] - \tag{55}$$

$$\frac{1}{N^2} \operatorname*{\mathbb{E}}_B \left[ \sum_{x^{(n)} \in B} \sum_{x^{(m)} \in B \setminus \{x^{(n)}\}} \nabla w_{ij}|x^{(n)} \nabla w_{ij}|x^{(m)} \right] \tag{56}$$

$$= \frac{1}{N^2} \operatorname*{\mathbb{E}}_B \left[ \sum_{x^{(n)} \in B} \sum_{x^{(m)} \in B \setminus \{x^{(n)}\}} \operatorname*{\mathbb{E}}_v \left[ \nabla w_{ij}|x^{(n)} \nabla w_{ij}|x^{(m)} v_{ij}^4 + \right. \right. \tag{57}$$

$$\nabla w_{ij}|x^{(n)} v_{ij}^2 \sum_{i'j' \neq ij} \nabla w_{i'j'}|x^{(m)} v_{i'j'} v_{ij} + \nabla w_{ij}|x^{(m)} v_{ij}^2 \sum_{i'j' \neq ij} \nabla w_{i'j'}|x^{(n)} v_{i'j'} v_{ij} + \tag{58}$$

$$\left. \left. \sum_{i'j' \neq ij} \nabla w_{i'j'}|x^{(n)} v_{i'j'} v_{ij} \sum_{i'j' \neq ij} \nabla w_{i'j'}|x^{(m)} v_{i'j'} v_{ij} \right] \right] - \tag{59}$$

$$\frac{1}{N^2} \left( \operatorname*{\mathbb{E}}_{x^{(n)}} \operatorname*{\mathbb{E}}_{x^{(m)}} \left[ \sum_n \sum_{m \neq n} \nabla w_{ij}|x^{(n)} \nabla w_{ij}|x^{(m)} \right] \right) \tag{60}$$

$$= \frac{1}{N^2} \mathbb{E}_B \left[ \sum_{x^{(n)} \in B} \sum_{x^{(m)} \in B \setminus \{x^{(n)}\}} \mathbb{E}_v \left[ \nabla w_{ij} | x^{(n)} \nabla w_{ij} | x^{(m)} v_{ij}^4 + \right. \right. \tag{61}$$

$$\left. \left. \left( \sum_{i'j' \neq ij} \nabla w_{i'j'} | x^{(n)} v_{i'j'} \right) \left( \sum_{i'j' \neq ij} \nabla w_{i'j'} | x^{(m)} v_{i'j'} \right) v_{ij}^2 \right] \right] - \frac{1}{N^2} \left( \sum_n \sum_{m \neq n} \overline{\nabla w}_{ij}^2 \right) \tag{62}$$

$$= \frac{1}{N^2} \mathbb{E}_B \left[ \sum_{x^{(n)} \in B} \sum_{x^{(m)} \in B \setminus \{x^{(n)}\}} \mathbb{E}_v \left[ \nabla w_{ij} | x^{(n)} \nabla w_{ij} | x^{(m)} v_{ij}^4 + \right. \right. \tag{63}$$

$$\left. \left. \sum_{i'j' \neq ij} \nabla w_{i'j'} | x^{(n)} \nabla w_{i'j'} | x^{(m)} v_{i'j'}^2 v_{ij}^2 + \sum_{i'j' \neq ij} \sum_{i''j'' \neq ij, i'j'} \nabla w_{i'j'} | x^{(n)} \nabla w_{i'j'} | x^{(m)} v_{i'j'} v_{i''j''} v_{ij}^2 \right] \right] - \tag{64}$$

$$\frac{1}{N^2} \left( \sum_n \sum_{m \neq n} \overline{\nabla w}_{ij}^2 \right) \tag{65}$$

$$= \frac{1}{N^2} \mathbb{E}_B \left[ \sum_{x^{(n)} \in B} \sum_{x^{(m)} \in B \setminus \{x^{(n)}\}} \mathbb{E}_v \left[ \nabla w_{ij} | x^{(n)} \nabla w_{ij} | x^{(m)} v_{ij}^4 + \sum_{i'j' \neq ij} \nabla w_{i'j'} | x^{(n)} \nabla w_{i'j'} | x^{(m)} v_{i'j'}^2 v_{ij}^2 \right] \right] - \tag{66}$$

$$\frac{1}{N^2} \left( \sum_n \sum_{m \neq n} \overline{\nabla w}_{ij}^2 \right) \tag{67}$$

$$= \frac{1}{N^2} \mathbb{E}_B \left[ \sum_{x^{(n)} \in B} \sum_{x^{(m)} \in B \setminus \{x^{(n)}\}} \nabla w_{ij} | x^{(n)} \nabla w_{ij} | x^{(m)} \mathbb{E}_v \left[ v_{ij}^4 \right] + \right. \tag{68}$$

$$\left. \sum_{i'j' \neq ij} \nabla w_{i'j'} | x^{(n)} \nabla w_{i'j'} | x^{(m)} \mathbb{E}_v \left[ v_{i'j'}^2 \right] \mathbb{E}_v \left[ v_{ij}^2 \right] \right] - \frac{1}{N^2} \left( \sum_n \sum_{m \neq n} \overline{\nabla w}_{ij}^2 \right) \tag{69}$$

$$= \frac{1}{N^2} \mathbb{E}_B \left[ \sum_{x^{(n)} \in B} \sum_{x^{(m)} \in B \setminus \{x^{(n)}\}} 3 \nabla w_{ij} | x^{(n)} \nabla w_{ij} | x^{(m)} + \sum_{i'j' \neq ij} \nabla w_{i'j'} | x^{(n)} \nabla w_{i'j'} | x^{(m)} \right] - \tag{70}$$

$$\frac{1}{N^2} \left( \sum_n \sum_{m \neq n} \overline{\nabla w}_{ij}^2 \right) \tag{71}$$

$$= \frac{1}{N^2} \left[ \sum_n \sum_{m \neq n} \left( 3 \mathbb{E}_x \left[ \nabla w_{ij} | x \right]^2 + \sum_{i'j' \neq ij} \mathbb{E}_x \left[ \nabla w_{i'j'} | x \right]^2 \right) \right] - \frac{1}{N^2} \left( \sum_n \sum_{m \neq n} \overline{\nabla w}_{ij}^2 \right) \tag{72}$$

$$= \frac{1}{N^2} \left[ \sum_n \sum_{m \neq n} \left( 3 \overline{\nabla w}_{ij}^2 + \sum_{i'j' \neq ij} \overline{\nabla w}_{i'j'}^2 \right) \right] - \frac{1}{N^2} \left( \sum_n \sum_{m \neq n} \overline{\nabla w}_{ij}^2 \right) \tag{73}$$

$$= \frac{1}{N^2} \sum_n \sum_{m \neq n} \left( 2 \overline{\nabla w}_{ij}^2 + \sum_{i'j' \neq ij} \overline{\nabla w}_{i'j'}^2 \right) = \frac{N(N-1)}{N^2} \left( \overline{\nabla w}_{ij}^2 + \sum_{i'j'} \overline{\nabla w}_{i'j'}^2 \right). \tag{74}$$

Lastly, we average the variance across all weight dimensions:

$$
\mathrm{mVar}(g_w(w_{ij})) = \frac{1}{pq} \sum_{ij} \mathrm{Var}(g_w(w_{ij})) \tag{75}
$$

$$
= \frac{1}{pq} \sum_{ij} \{Z_1 + Z_2 + Z_3\} \tag{76}
$$

$$
= \frac{1}{pq} \sum_{ij} \left\{ \frac{1}{N} \operatorname*{Var}_x \left( \nabla w_{ij}|_x \right) + \right. \tag{77}
$$

$$
\frac{1}{N} \left[ \overline{\nabla w}_{ij}^2 + \operatorname*{Var}_x (\nabla w_{ij}|x) + \sum_{i'j'} \left( \overline{\nabla w}_{i'j'}^2 + \operatorname*{Var}_x (\nabla w_{i'j'}|x) \right) \right] + \tag{78}
$$

$$
\frac{N(N-1)}{N^2} \left( \overline{\nabla w}_{ij}^2 + \sum_{i'j'} \overline{\nabla w}_{i'j'}^2 \right) \right\} \tag{79}
$$

$$
= \frac{1}{pq} \sum_{ij} \left\{ \frac{1}{N} \operatorname*{Var}_x \left( \nabla w_{ij}|_x \right) + \right. \tag{80}
$$

$$
\frac{1}{N} \left[ \operatorname*{Var}_x (\nabla w_{ij}|x) + \sum_{i'j'} \operatorname*{Var}_x (\nabla w_{i'j'}|x) \right] + \left( \overline{\nabla w}_{ij}^2 + \sum_{i'j'} \overline{\nabla w}_{i'j'}^2 \right) \right\} \tag{81}
$$

$$
= \frac{2}{N} \mathrm{mVar}\left( \nabla w \right) + \frac{pq}{N} \mathrm{mVar}\left( \nabla w \right) + (pq+1)\,\mathrm{mSqNorm}(\overline{\nabla w}) \tag{82}
$$

$$
= \frac{pq+2}{N} V + (pq+1)S. \tag{83}
$$

If the perturbations are independent, we show that $Z_3$ is 0.

$$
Z_3 = \frac{1}{N^2} \operatorname*{\mathbb{E}}_B \left[ \sum_{x^{(n)} \in B} \sum_{x^{(m)} \in B \setminus \{x^{(n)}\}} \operatorname*{Cov}_v \left( g_w(w_{ij})| \, x^{(n)}, g_w(w_{ij})| \, x^{(m)} \right) \right] \tag{84}
$$

$$
= \frac{1}{N^2} \operatorname*{\mathbb{E}}_B \left[ \sum_{x^{(n)} \in B} \sum_{x^{(m)} \in B \setminus \{x^{(n)}\}} \operatorname*{\mathbb{E}}_v \left[ g_w(w_{ij})| \, x^{(n)} \; g_w(w_{ij})| \, x^{(m)} \right] - \operatorname*{\mathbb{E}}_v \left[ g_w(w_{ij})| \, x^{(n)} \right] \operatorname*{\mathbb{E}}_v \left[ g_w(w_{ij})| \, x^{(m)} \right] \right] \tag{85}
$$

$$
= \frac{1}{N^2} \operatorname*{\mathbb{E}}_B \left[ \sum_{x^{(n)} \in B} \sum_{x^{(m)} \in B \setminus \{x^{(n)}\}} \operatorname*{\mathbb{E}}_v \left[ g_w(w_{ij})| \, x^{(n)} \; g_w(w_{ij})| \, x^{(m)} \right] - \nabla w_{ij}|x^{(n)} \nabla w_{ij}|x^{(m)} \right] \tag{86}
$$

$$
= \frac{1}{N^2} \operatorname*{\mathbb{E}}_B \left[ \sum_{x^{(n)} \in B} \sum_{x^{(m)} \in B \setminus \{x^{(n)}\}} \operatorname*{\mathbb{E}}_v \left[ \left( \sum_{j'} \nabla w_{ij'}|x^{(n)} v_{i'j'}^{(n)} \right) v_{ij}^{(n)} \left( \sum_{j'} \nabla w_{ij'}|x^{(m)} v_{i'j'}^{(m)} \right) v_{ij}^{(m)} \right] \right. \tag{87}
$$

$$
\left. - \nabla w_{ij}|x^{(n)} \nabla w_{ij}|x^{(m)} \right] \tag{88}
$$

$$
= \frac{1}{N^2} \operatorname*{\mathbb{E}}_B \left[ \sum_{x^{(n)} \in B} \sum_{x^{(m)} \in B \setminus \{x^{(n)}\}} \operatorname*{\mathbb{E}}_v \left[ \sum_{j'} \sum_{j''} \nabla w_{ij'}|x^{(n)} \nabla w_{ij''}|x^{(m)} v_{i'j'}^{(n)} v_{i''j''}^{(m)} v_{ij}^{(n)} v_{ij}^{(m)} \right] \right] - \tag{89}
$$

$$
\frac{1}{N^2} \operatorname*{\mathbb{E}}_B \left[ \sum_{x^{(n)} \in B} \sum_{x^{(m)} \in B \setminus \{x^{(n)}\}} \nabla w_{ij}|x^{(n)} \nabla w_{ij}|x^{(m)} \right] \tag{90}
$$

$$= \frac{1}{N^2} \mathop{\mathbb{E}}_{B} \left[ \sum_{x^{(n)} \in B} \sum_{x^{(m)} \in B \setminus \{x^{(n)}\}} \mathop{\mathbb{E}}_{v} \left[ \nabla w_{ij} | x^{(n)} \nabla w_{ij} | x^{(m)} v_{ij}^{(n)2} v_{ij}^{(m)2} + \right. \right. \tag{91}$$

$$\nabla w_{ij} | x^{(n)} v_{ij}^{(n)2} v_{ij}^{(m)} \sum_{i'j' \neq j} \nabla w_{ij'} | x^{(m)} v_{i'j'}^{(m)} + \nabla w_{ij} | x^{(m)} v_{ij}^{(m)2} v_{ij}^{(n)} \sum_{i'j' \neq j} \nabla w_{ij'} | x^{(n)} v_{i'j'}^{(n)} + \tag{92}$$

$$\sum_{i'j' \neq ij} \nabla w_{ij'} | x^{(m)} \nabla w_{ij'} | x^{(n)} v_{i'j'}^{(m)} v_{i'j'}^{(n)} v_{ij}^{(m)} v_{ij}^{(n)} + \tag{93}$$

$$\sum_{i'j' \neq ij} \sum_{i''j'' \notin \{ij, j'j'\}} \nabla w_{i'j'} | x^{(n)} v_{i'j'} \nabla w_{ij'} | x^{(m)} v_{i'j'}^{(n)} v_{i''j''}^{(m)} v_{ij}^{(n)} v_{ij}^{(m)} \left. \left. \right] \right] - \tag{94}$$

$$\frac{1}{N^2} \left( \mathop{\mathbb{E}}_{x^{(n)}} \mathop{\mathbb{E}}_{x^{(m)}} \left[ \sum_{n} \sum_{m \neq n} \nabla w_{ij} | x^{(n)} \nabla w_{ij} | x^{(m)} \right] \right) \tag{95}$$

$$= \frac{1}{N^2} \mathop{\mathbb{E}}_{B} \left[ \sum_{x^{(n)} \in B} \sum_{x^{(m)} \in B \setminus \{x^{(n)}\}} \mathop{\mathbb{E}}_{v} \left[ \nabla w_{ij} | x^{(n)} \nabla w_{ij} | x^{(m)} v_{ij}^{(n)2} v_{ij}^{(m)2} + \right. \right. \tag{96}$$

$$\sum_{i'j' \neq ij} \nabla w_{i'j'} | x^{(m)} \nabla w_{i'j'} | x^{(n)} v_{i'j'}^{(m)} v_{i'j'}^{(n)} v_{ij}^{(m)} v_{ij}^{(n)} \left. \left. \right] \right] - \tag{97}$$

$$\frac{1}{N^2} \left( \mathop{\mathbb{E}}_{x^{(n)}} \mathop{\mathbb{E}}_{x^{(m)}} \left[ \sum_{n} \sum_{m \neq n} \nabla w_{ij} | x^{(n)} \nabla w_{ij} | x^{(m)} \right] \right) \tag{98}$$

$$= \frac{1}{N^2} \mathop{\mathbb{E}}_{B} \left[ \sum_{x^{(n)} \in B} \sum_{x^{(m)} \in B \setminus \{x^{(n)}\}} \mathop{\mathbb{E}}_{v} \left[ \nabla w_{ij} | x^{(n)} \nabla w_{ij} | x^{(m)} v_{ij}^{(n)2} v_{ij}^{(m)2} \right] \right] - \tag{99}$$

$$\frac{1}{N^2} \left( \sum_{n} \sum_{m \neq n} \overline{\nabla w}_{ij}^2 \right) \tag{100}$$

$$= \frac{1}{N^2} \mathop{\mathbb{E}}_{B} \left[ \sum_{x^{(n)} \in B} \sum_{x^{(m)} \in B \setminus \{x^{(n)}\}} \nabla w_{ij} | x^{(n)} \nabla w_{ij} | x^{(m)} \mathop{\mathbb{E}}_{v} \left[ v_{ij}^{(n)2} \right] \mathop{\mathbb{E}}_{v} \left[ v_{ij}^{(m)2} \right] \right] - \tag{101}$$

$$\frac{1}{N^2} \left( \sum_{n} \sum_{m \neq n} \overline{\nabla w}_{ij}^2 \right) \tag{102}$$

$$= \frac{1}{N^2} \mathop{\mathbb{E}}_{B} \left[ \sum_{x^{(n)} \in B} \sum_{x^{(m)} \in B \setminus \{x^{(n)}\}} \nabla w_{ij} | x^{(n)} \nabla w_{ij} | x^{(m)} \right] - \frac{1}{N^2} \left( \sum_{n} \sum_{m \neq n} \overline{\nabla w}_{ij}^2 \right) \tag{103}$$

$$= \frac{1}{N^2} \left( \sum_{n} \sum_{m \neq n} \mathop{\mathbb{E}}_{x} [\nabla w_{ij} | x]^2 \right) - \frac{1}{N^2} \left( \sum_{n} \sum_{m \neq n} \overline{\nabla w}_{ij}^2 \right) \tag{104}$$

$$= 0. \tag{105}$$

Then the average variance becomes:

$$\mathrm{mVar}(g_w(w_{ij})) = \frac{1}{pq} \sum_{ij} \mathrm{Var}(g_w(w_{ij})) \tag{106}$$

$$= \frac{1}{pq} \sum_{ij} \{Z_1 + Z_2 + Z_3\} \tag{107}$$

$$= \frac{1}{pq} \sum_{ij} \left\{ \frac{1}{N} \mathrm{Var}_x \left( \nabla w_{ij} | x \right) + \right. \tag{108}$$

$$\frac{1}{N} \left[ \overline{\nabla w}_{ij}^2 + \mathrm{Var}_x(\nabla w_{ij} | x) + \sum_{i'j'} \left( \overline{\nabla w}_{i'j'}^2 + \mathrm{Var}_x(\nabla w_{i'j'} | x) \right) \right] \tag{109}$$

$$= \frac{pq+2}{N} \mathrm{mVar}(\nabla w) + \frac{pq+1}{N} \mathrm{mSqNorm}(\overline{\nabla w}) \tag{110}$$

$$= \frac{pq+2}{N} V + \frac{pq+1}{N} S. \tag{111}$$

$\square$

**Proposition 4.** *Let $p \times q$ be the size of the weight matrix, the element-wise average variance of the activity perturbed gradient estimator with a batch size $N$ is $\frac{q+2}{N} V + (q+1)S$ if the perturbations are shared across the batch, and $\frac{q+2}{N} V + \frac{q+1}{N} S$ if they are independent, where $V$ is the element-wise average variance of the true gradient, and $S$ is the element-wise average squared gradient.*

*Proof.*

$$Z_2 = \frac{1}{N} \mathbb{E}_x \left[ \mathrm{Var}_u \left( g_a(w_{ij}) | x \right) \right] \tag{112}$$

$$= \frac{1}{N} \mathbb{E}_x \left[ \mathrm{Var}_u \left( \left( \sum_{j'} \nabla w_{ij'} u_{j'} \right) u_j \right) \right] \tag{113}$$

$$= \frac{1}{N} \mathbb{E}_x \left[ \mathrm{Var}_u \left( \nabla w_{ij} u_j^2 + \sum_{j' \neq j} \nabla w_{j'} u_j u_{j'} \right) \right] \tag{114}$$

$$= \frac{1}{N} \mathbb{E}_x \left[ \mathrm{Var}_u \left( \nabla w_{ij} u_j^2 \right) + \mathrm{Var}_u \left( \sum_{j' \neq j} \nabla w_{ij'} u_j u_{j'} \right) + \right. \tag{115}$$

$$2 \mathrm{Cov}_u \left( \nabla w_{ij} u_j^2, \sum_{j' \neq j} \nabla w_{ij} u_j u_{j'} \right) \right] \tag{116}$$

$$= \frac{1}{N} \mathbb{E}_x \left[ \mathrm{Var}_u \left( \nabla w_{ij} u_j^2 \right) + \mathrm{Var}_u \left( \sum_{i'j' \neq ij} \nabla w_{ij'} u_j u_{j'} \right) + \right. \tag{117}$$

$$2 \mathbb{E}_u \left[ \sum_{j' \neq j} \nabla w_{ij} \nabla w_{ij'} u_j^3 u_{j'} \right] - 2 \mathbb{E}_u \left[ \nabla w_{ij} u_j^2 \right] \mathbb{E}_u \left[ \sum_{j' \neq j} \nabla w_{ij'} u_j u_{j'} \right] \right] \tag{118}$$

$$= \frac{1}{N} \mathbb{E}_x \left[ \nabla w_{ij}^2 \mathrm{Var}_u \left( u_j^2 \right) + \mathrm{Var}_u \left( \sum_{j' \neq j} \nabla w_{ij'} u_j u_{j'} \right) + \right. \tag{119}$$

$$2 \sum_{j' \neq j} \nabla w_{ij} \nabla w_{ij'} \mathbb{E}_u \left[ u_j^3 u_{j'} \right] - 2 \nabla w_{ij} \mathbb{E}_u \left[ u_j^2 \right] \left( \sum_{j' \neq j} \nabla w_{ij'} \mathbb{E}_u \left[ u_j u_{j'} \right] \right) \right] \tag{120}$$

$$= \frac{1}{N} \mathop{\mathbb{E}}_{x} \left[ \nabla w_{ij}^2 \mathop{\mathrm{Var}}_u \left( u_j^2 \right) + \mathop{\mathrm{Var}}_u \left( \sum_{j' \neq j} \nabla w_{ij'} u_j u_{j'} \right) + \right. \tag{121}$$

$$\left. 2 \sum_{j' \neq j} \nabla w_{ij} \nabla w_{ij'} \cdot 0 - 2 \nabla w_{ij} \cdot 1 \left( \sum_{j' \neq j} \nabla w_{i'j'} \cdot 0 \right) \right] \tag{122}$$

$$= \frac{1}{N} \mathop{\mathbb{E}}_{x} \left[ \nabla w_{ij}^2 \mathop{\mathrm{Var}}_u \left( u_j^2 \right) + \mathop{\mathrm{Var}}_u \left( \sum_{j' \neq j} \nabla w_{ij'} u_j u_{j'} \right) \right] \tag{123}$$

$$= \frac{1}{N} \mathop{\mathbb{E}}_{x} \left[ \nabla w_{ij}^2 \cdot (\mathop{\mathbb{E}}_u [u_j^4] - \mathop{\mathbb{E}}_u [u_j^2]^2) + \sum_{j' \neq j} \mathop{\mathrm{Var}}_u (\nabla w_{ij'} u_j u_{j'}) \right] \tag{124}$$

$$= \frac{1}{N} \mathop{\mathbb{E}}_{x} \left[ \nabla w_{ij}^2 (3 \mathop{\mathrm{Var}}_u (u_j)^2 - \mathop{\mathbb{E}}_u [u_j^2]^2) + \sum_{j' \neq j} \nabla w_{j'}^2 \mathop{\mathrm{Var}}_u (u_j u_{j'}) \right] \tag{125}$$

$$= \frac{1}{N} \mathop{\mathbb{E}}_{x} \left[ 2 \nabla w_{ij}^2 \right] + \tag{126}$$

$$\frac{1}{N} \mathop{\mathbb{E}}_{x} \left[ \sum_{j' \neq j} \nabla w_{ij'}^2 (\mathop{\mathrm{Var}}_u [u_j] + \mathop{\mathbb{E}}_u [u_{j'}]^2)(\mathop{\mathrm{Var}}_u [u_{j'}] + \mathop{\mathbb{E}}_u [u_{j'}]^2) - \mathop{\mathbb{E}}_u [u_j]^2 \mathop{\mathbb{E}}_u [u_{j'}]^2 \right] \tag{127}$$

$$= \frac{2}{N} \overline{\nabla w}_{ij}^2 + \frac{2}{N} \mathop{\mathrm{Var}}_x (\nabla w_{ij} | x) + \frac{1}{N} \mathop{\mathbb{E}}_{x} \left[ \sum_{j' \neq j} \nabla w_{ij'}^2 \mathop{\mathrm{Var}}_u (u_j) \mathop{\mathrm{Var}}_u (u_{j'}) \right] \tag{128}$$

$$= \frac{2}{N} \overline{\nabla w}_{ij}^2 + \frac{2}{N} \mathop{\mathrm{Var}}_x (\nabla w_{ij} | x) + \frac{1}{N} \sum_{j' \neq j} \mathop{\mathbb{E}}_{x} \left[ \nabla w_{j'}^2 \right] \tag{129}$$

$$= \frac{1}{N} \left[ \overline{\nabla w}_{ij}^2 + \mathop{\mathrm{Var}}_x (\nabla w_{ij} | x) + \sum_{j'} \left( \overline{\nabla w}_{ij'}^2 + \mathop{\mathrm{Var}}_x (\nabla w_{ij'} | x) \right) \right]. \tag{130}$$

$Z_3$ is nonzero if the perturbations are shared within a batch. Assuming that the perturbations are shared,

$$Z_3 = \frac{1}{N^2} \mathop{\mathbb{E}}_{B} \left[ \sum_{x^{(n)} \in B} \sum_{x^{(m)} \in B \setminus \{x^{(n)}\}} \mathop{\mathrm{Cov}}_u (g_a(w_{ij}) | x^{(n)}, g_a(w_{ij}) | x^{(m)}) \right] \tag{131}$$

$$= \frac{1}{N^2} \mathop{\mathbb{E}}_{B} \left[ \sum_{x^{(n)} \in B} \sum_{x^{(m)} \in B \setminus \{x^{(n)}\}} \mathop{\mathbb{E}}_u \left[ g_a(w_{ij}) | x^{(n)} g_a(w_{ij}) | x^{(m)} \right] - \mathop{\mathbb{E}}_u \left[ g_a(w_{ij}) | x^{(n)} \right] \mathop{\mathbb{E}}_u \left[ g_a(w_{ij}) | x^{(m)} \right] \right] \tag{132}$$

$$= \frac{1}{N^2} \mathop{\mathbb{E}}_{B} \left[ \sum_{x^{(n)} \in B} \sum_{x^{(m)} \in B \setminus \{x^{(n)}\}} \mathop{\mathbb{E}}_u \left[ g_a(w_{ij}) | x^{(n)} g_a(w_{ij}) | x^{(m)} \right] - \nabla w_{ij} | x^{(n)} \nabla w_{ij} | x^{(m)} \right] \tag{133}$$

$$= \frac{1}{N^2} \mathop{\mathbb{E}}_{B} \left[ \sum_{x^{(n)} \in B} \sum_{x^{(m)} \in B \setminus \{x^{(n)}\}} \mathop{\mathbb{E}}_u \left[ \left( \sum_{j'} \nabla w_{ij'} | x^{(n)} u_{j'} \right) u_j \left( \sum_{j'} \nabla w_{ij'} | x^{(m)} u_{j'} \right) u_j \right] - \right. \tag{134}$$

$$\left. \nabla w_{ij} | x^{(n)} \nabla w_{ij} | x^{(m)} \right] \tag{135}$$

$$= \frac{1}{N^2} \mathbb{E}_B \left[ \sum_{x^{(n)} \in B} \sum_{x^{(m)} \in B \setminus \{x^{(n)}\}} \mathbb{E}_u \left[ \sum_{j'} \sum_{j''} \nabla w_{ij'} | x^{(n)} \nabla w_{ij''} | x^{(m)} u_{j'} u_{j''} u_j^2 \right] \right] - \tag{136}$$

$$\frac{1}{N^2} \mathbb{E}_B \left[ \sum_{x^{(n)} \in B} \sum_{x^{(m)} \in B \setminus \{x^{(n)}\}} \nabla w_{ij} | x^{(n)} \nabla w_{ij} | x^{(m)} \right] \tag{137}$$

$$= \frac{1}{N^2} \mathbb{E}_B \left[ \sum_{x^{(n)} \in B} \sum_{x^{(m)} \in B \setminus \{x^{(n)}\}} \mathbb{E}_u \left[ \nabla w_{ij} | x^{(n)} \nabla w_{ij} | x^{(m)} v_{ij}^4 + \right. \right. \tag{138}$$

$$\nabla w_{ij} | x^{(n)} u_j^3 \sum_{j' \neq j} \nabla w_{ij'} | x^{(m)} u_{j'} + \nabla w_{ij} | x^{(m)} u_j^3 \sum_{j' \neq j} \nabla w_{ij'} | x^{(n)} u_{j'} + \tag{139}$$

$$\sum_{j' \neq j} \nabla w_{ij'} | x^{(m)} \nabla w_{ij'} | x^{(n)} u_{j'}^2 u_j^2 + \tag{140}$$

$$\sum_{j' \neq j} \sum_{j'' \notin \{j,j'\}} \nabla w_{ij'} | x^{(n)} u_{j'} \nabla w_{ij'} | x^{(m)} u_{j'} u_{j''} u_j^2 \right] \right] - \tag{141}$$

$$\frac{1}{N^2} \left( \mathbb{E}_{x^{(n)}} \mathbb{E}_{x^{(m)}} \left[ \sum_n \sum_{m \neq n} \nabla w_{ij} | x^{(n)} \nabla w_{ij} | x^{(m)} \right] \right) \tag{142}$$

$$= \frac{1}{N^2} \mathbb{E}_B \left[ \sum_{x^{(n)} \in B} \sum_{x^{(m)} \in B \setminus \{x^{(n)}\}} \mathbb{E}_u \left[ \nabla w_{ij} | x^{(n)} \nabla w_{ij} | x^{(m)} v_{ij}^4 + \sum_{j' \neq j} \nabla w_{ij'} | x^{(m)} \nabla w_{ij'} | x^{(n)} u_{j'}^2 u_j^2 \right] \right] - \tag{143}$$

$$\frac{1}{N^2} \left( \mathbb{E}_{x^{(n)}} \mathbb{E}_{x^{(m)}} \left[ \sum_n \sum_{m \neq n} \nabla w_{ij} | x^{(n)} \nabla w_{ij} | x^{(m)} \right] \right) \tag{144}$$

$$= \frac{1}{N^2} \mathbb{E}_B \left[ \sum_{x^{(n)} \in B} \sum_{x^{(m)} \in B \setminus \{x^{(n)}\}} \mathbb{E}_u \left[ \nabla w_{ij} | x^{(n)} \nabla w_{ij} | x^{(m)} v_{ij}^4 + \sum_{j' \neq j} \nabla w_{ij'} | x^{(m)} \nabla w_{ij'} | x^{(n)} u_{j'}^2 u_j^2 \right] \right] - \tag{145}$$

$$\frac{1}{N^2} \left( \sum_n \sum_{m \neq n} \overline{\nabla w}_{ij}^2 \right) \tag{146}$$

$$= \frac{1}{N^2} \mathbb{E}_B \left[ \sum_{x^{(n)} \in B} \sum_{x^{(m)} \in B \setminus \{x^{(n)}\}} \nabla w_{ij} | x^{(n)} \nabla w_{ij} | x^{(m)} \mathbb{E}_v \left[ v_{ij}^4 \right] + \sum_{j' \neq j} \nabla w_{ij'} | x^{(n)} \nabla w_{ij'} | x^{(m)} \mathbb{E}_u \left[ u_{j'}^2 \right] \mathbb{E}_u \left[ u_j^2 \right] \right] - \tag{147}$$

$$\frac{1}{N^2} \left( \sum_n \sum_{m \neq n} \overline{\nabla w}_{ij}^2 \right) \tag{148}$$

$$= \frac{1}{N^2} \mathbb{E}_B \left[ \sum_{x^{(n)} \in B} \sum_{x^{(m)} \in B \setminus \{x^{(n)}\}} 3 \nabla w_{ij} | x^{(n)} \nabla w_{ij} | x^{(m)} + \sum_{j' \neq j} \nabla w_{ij'} | x^{(n)} \nabla w_{ij'} | x^{(m)} \right] - \tag{149}$$

$$\frac{1}{N^2} \left( \sum_n \sum_{m \neq n} \overline{\nabla w}_{ij}^2 \right) \tag{150}$$

$$= \frac{1}{N^2} \left[ \sum_n \sum_{m \neq n} \left( 3 \, \mathbb{E}_x \left[ \nabla w_{ij} | x \right]^2 + \sum_{j' \neq j} \mathbb{E}_x \left[ \nabla w_{ij'} | x \right]^2 \right) \right] - \frac{1}{N^2} \left( \sum_n \sum_{m \neq n} \overline{\nabla w}_{ij}^2 \right) \tag{151}$$

$$= \frac{1}{N^2} \left[ \sum_n \sum_{m \neq n} \left( 2 \overline{\nabla w}_{ij}^2 + \sum_{j' \neq j} \overline{\nabla w}_{ij'}^2 \right) \right] - \frac{1}{N^2} \left( \sum_n \sum_{m \neq n} \overline{\nabla w}_{ij}^2 \right) \tag{152}$$

$$= \frac{1}{N^2} \left[ \sum_n \sum_{m \neq n} \left( \overline{\nabla w}_{ij}^2 + \sum_{j' \neq j} \overline{\nabla w}_{ij'}^2 \right) \right] \tag{153}$$

$$= \frac{N(N-1)}{N^2} \left( \overline{\nabla w}_{ij}^2 + \sum_{j' \neq j} \overline{\nabla w}_{ij'}^2 \right). \tag{154}$$

Then we compute the average variance across all weight dimensions (for shared perturbation):

$$\mathrm{mVar}(g_a(w_{ij})) = \frac{1}{pq} \sum_{ij} \mathrm{Var}(g_a(w_{ij})) \tag{155}$$

$$= \frac{1}{pq} \sum_{ij} \{ Z_1 + Z_2 + Z_3 \} \tag{156}$$

$$= \frac{1}{pq} \sum_{ij} \left\{ \frac{1}{N} \, \mathrm{Var}_x \left( \nabla w_{ij} | x \right) + \right. \tag{157}$$

$$\frac{1}{N} \left[ \overline{\nabla w}_{ij}^2 + \mathrm{Var}_x (\nabla w_{ij} | x) + \sum_{j'} \left( \overline{\nabla w}_{ij'}^2 + \mathrm{Var}_x (\nabla w_{ij'} | x) \right) \right] + \tag{158}$$

$$\frac{N(N-1)}{N^2} \left( \overline{\nabla w}_{ij}^2 + \sum_{j' \neq j} \overline{\nabla w}_{ij'}^2 \right) \tag{159}$$

$$= \frac{1}{pq} \sum_{ij} \left\{ \frac{1}{N} \, \mathrm{Var}_x \left( \nabla w_{ij} | x \right) + \right. \tag{160}$$

$$\left. \frac{1}{N} \left[ \mathrm{Var}_x (\nabla w_{ij} | x) + \sum_{j'} \mathrm{Var}_x (\nabla w_{ij'} | x) \right] + \left( \overline{\nabla w}_{ij}^2 + \sum_{j'} \overline{\nabla w}_{ij'}^2 \right) \right\} \tag{161}$$

$$= \frac{2}{N} \, \mathrm{mVar} \left( \nabla w \right) + \frac{q}{N} \, \mathrm{mVar} \left( \nabla w \right) + (q+1) \, \mathrm{mSqNorm}(\overline{\nabla w}) \tag{162}$$

$$= \frac{q+2}{N} V + (q+1) S. \tag{163}$$

If the perturbations are independent, we show that $Z_3$ is 0.

$$Z_3 = \frac{1}{N^2} \mathbb{E}_B \left[ \sum_{x^{(n)} \in B} \sum_{x^{(m)} \in B \setminus \{x^{(n)}\}} \mathrm{Cov}_u ( g_a(w_{ij}) | x^{(n)}, g_a(w_{ij}) | x^{(m)} ) \right] \tag{164}$$

$$= \frac{1}{N^2} \mathbb{E}_B \left[ \sum_{x^{(n)} \in B} \sum_{x^{(m)} \in B \setminus \{x^{(n)}\}} \mathbb{E}_u \left[ g_a(w_{ij}) | x^{(n)} \, g_a(w_{ij}) | x^{(m)} \right] - \mathbb{E}_u \left[ g_a(w_{ij}) | x^{(n)} \right] \mathbb{E}_u \left[ g_a(w_{ij}) | x^{(m)} \right] \right] \tag{165}$$

$$= \frac{1}{N^2} \mathbb{E}_B \left[ \sum_{x^{(n)} \in B} \sum_{x^{(m)} \in B \setminus \{x^{(n)}\}} \mathbb{E}_u \left[ g_a(w_{ij}) | \, x^{(n)} \, g_a(w_{ij}) | \, x^{(m)} \right] - \nabla w_{ij} | x^{(n)} \nabla w_{ij} | x^{(m)} \right] \quad (166)$$

$$= \frac{1}{N^2} \mathbb{E}_B \left[ \sum_{x^{(n)} \in B} \sum_{x^{(m)} \in B \setminus \{x^{(n)}\}} \mathbb{E}_u \left[ \left( \sum_{j'} \nabla w_{ij'} | x^{(n)} u_{j'}^{(n)} \right) u_j^{(n)} \left( \sum_{j'} \nabla w_{ij'} | x^{(m)} u_{j'}^{(m)} \right) u_j^{(m)} \right] - \right.$$
$$\quad (167)$$

$$\left. \nabla w_{ij} | x^{(n)} \nabla w_{ij} | x^{(m)} \right] \quad (168)$$

$$= \frac{1}{N^2} \mathbb{E}_B \left[ \sum_{x^{(n)} \in B} \sum_{x^{(m)} \in B \setminus \{x^{(n)}\}} \mathbb{E}_u \left[ \sum_{j'} \sum_{j''} \nabla w_{ij'} | x^{(n)} \nabla w_{ij''} | x^{(m)} u_{j'}^{(n)} u_{j''}^{(m)} u_j^{(n)} u_j^{(m)} \right] \right] - \quad (169)$$

$$\frac{1}{N^2} \mathbb{E}_B \left[ \sum_{x^{(n)} \in B} \sum_{x^{(m)} \in B \setminus \{x^{(n)}\}} \nabla w_{ij} | x^{(n)} \nabla w_{ij} | x^{(m)} \right] \quad (170)$$

$$= \frac{1}{N^2} \mathbb{E}_B \left[ \sum_{x^{(n)} \in B} \sum_{x^{(m)} \in B \setminus \{x^{(n)}\}} \mathbb{E}_u \left[ \nabla w_{ij} | x^{(n)} \nabla w_{ij} | x^{(m)} u_j^{(n)2} u_j^{(m)2} + \right. \right. \quad (171)$$

$$\nabla w_{ij} | x^{(n)} u_j^{(n)2} u_j^{(m)} \sum_{j' \neq j} \nabla w_{ij'} | x^{(m)} u_{j'}^{(m)} + \nabla w_{ij} | x^{(m)} u_j^{(m)2} u_j^{(n)} \sum_{j' \neq j} \nabla w_{ij'} | x^{(n)} u_{j'}^{(n)} + \quad (172)$$

$$\sum_{j' \neq j} \nabla w_{ij'} | x^{(m)} \nabla w_{ij'} | x^{(n)} u_{j'}^{(m)} u_{j'}^{(n)} u_j^{(m)} u_j^{(n)} + \quad (173)$$

$$\left. \left. \sum_{j' \neq j} \sum_{j'' \notin \{j, j'\}} \nabla w_{ij'} | x^{(n)} u_{j'} \nabla w_{ij'} | x^{(m)} u_{j'}^{(n)} u_{j''}^{(m)} u_j^{(n)} u_j^{(m)} \right] \right] - \quad (174)$$

$$\frac{1}{N^2} \left( \mathbb{E}_{x^{(n)}} \mathbb{E}_{x^{(m)}} \left[ \sum_n \sum_{m \neq n} \nabla w_{ij} | x^{(n)} \nabla w_{ij} | x^{(m)} \right] \right) \quad (175)$$

$$= \frac{1}{N^2} \mathbb{E}_B \left[ \sum_{x^{(n)} \in B} \sum_{x^{(m)} \in B \setminus \{x^{(n)}\}} \mathbb{E}_u \left[ \nabla w_{ij} | x^{(n)} \nabla w_{ij} | x^{(m)} u_j^{(n)2} u_j^{(m)2} + \right. \right. \quad (176)$$

$$\left. \left. \sum_{j' \neq j} \nabla w_{ij'} | x^{(m)} \nabla w_{ij'} | x^{(n)} u_{j'}^{(m)} u_{j'}^{(n)} u_j^{(m)} u_j^{(n)} \right] \right] - \quad (177)$$

$$\frac{1}{N^2} \left( \mathbb{E}_{x^{(n)}} \mathbb{E}_{x^{(m)}} \left[ \sum_n \sum_{m \neq n} \nabla w_{ij} | x^{(n)} \nabla w_{ij} | x^{(m)} \right] \right) \quad (178)$$

$$= \frac{1}{N^2} \mathbb{E}_B \left[ \sum_{x^{(n)} \in B} \sum_{x^{(m)} \in B \setminus \{x^{(n)}\}} \mathbb{E}_u \left[ \nabla w_{ij} | x^{(n)} \nabla w_{ij} | x^{(m)} u_j^{(n)2} u_j^{(m)2} \right] \right] - \quad (179)$$

$$\frac{1}{N^2} \left( \sum_n \sum_{m \neq n} \overline{\nabla w}_{ij}^2 \right) \quad (180)$$

$$= \frac{1}{N^2} \mathbb{E}_B \left[ \sum_{x^{(n)} \in B} \sum_{x^{(m)} \in B \setminus \{x^{(n)}\}} \nabla w_{ij} | x^{(n)} \nabla w_{ij} | x^{(m)} \mathbb{E}_u \left[ u_j^{(n)2} \right] \mathbb{E}_u \left[ u_j^{(m)2} \right] \right] - \quad (181)$$

$$\frac{1}{N^2} \left( \sum_n \sum_{m \neq n} \overline{\nabla w}_{ij}^2 \right) \quad (182)$$

$$= \frac{1}{N^2} \mathop{\mathbb{E}}_{B} \left[ \sum_{x^{(n)} \in B} \sum_{x^{(m)} \in B \setminus \{x^{(n)}\}} \nabla w_{ij} |x^{(n)} \nabla w_{ij} |x^{(m)} \right] - \frac{1}{N^2} \left( \sum_n \sum_{m \neq n} \overline{\nabla w}_{ij}^2 \right) \tag{183}$$

$$= \frac{1}{N^2} \left( \sum_n \sum_{m \neq n} \mathop{\mathbb{E}}_{x} [\nabla w_{ij} |x]^2 \right) - \frac{1}{N^2} \left( \sum_n \sum_{m \neq n} \overline{\nabla w}_{ij}^2 \right) \tag{184}$$

$$= 0. \tag{185}$$

Then the average variance becomes:

$$\mathrm{mVar}(g_a(w_{ij})) = \frac{1}{pq} \sum_{ij} \mathrm{Var}(g_a(w_{ij})) \tag{186}$$

$$= \frac{1}{pq} \sum_{ij} \{Z_1 + Z_2 + Z_3\} \tag{187}$$

$$= \frac{1}{pq} \sum_{ij} \left\{ \frac{1}{N} \mathop{\mathrm{Var}}_{x} (\nabla w_{ij} |_x) + \right. \tag{188}$$

$$\left. \frac{1}{N} \left[ \overline{\nabla w}_{ij}^2 + \mathop{\mathrm{Var}}_{x}(\nabla w_{ij} |x) + \sum_{j'} \left( \overline{\nabla w}_{ij'}^2 + \mathop{\mathrm{Var}}_{x}(\nabla w_{ij'} |x) \right) \right] \right\} \tag{189}$$

$$= \frac{q+2}{N} \mathrm{mVar} (\nabla w) + \frac{q+1}{N} \mathrm{mSqNorm}(\overline{\nabla w}) \tag{190}$$

$$= \frac{q+2}{N} V + \frac{q+1}{N} S. \tag{191}$$

$\square$

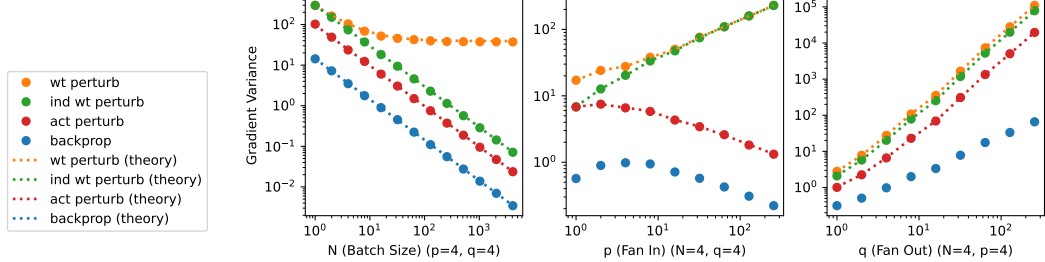

Figure 8: Numerical verification of the theoretical variance properties

## 10 NUMERICAL SIMULATION OF VARIANCES

In Figure 8, we ran numerical simulation experiments to verify our analytical variance properties. We used a multi-layer network with 4 input units, 4 hidden units, 1 output unit, a $\tanh$ activation function, and the mean squared error loss. We varied the batch size ($N$) between 1 and 4096. We tested the gradient estimator of the first layer weights using 5000 random samples. We also calculated the theoretical variance by applying the gradient norm and gradient variance constants found by backprop, from 5000 mini-batch true gradients. We then fixed the batch size to be 4 and vary the number of input units ($p$, fan in) and the number of hidden units ($q$, fan out) between 1 and 256. The theoretical variance for backprop was only computed for the batch size experiment since it is an inverse relationship ($\frac{1}{N}$), but for fan in and fan out, we do not aim to analyze the theoretical variances here. **"wt perturb"** stands for weight perturbation with shared noise; **"ind wt perturb"** stands for weight perturbation with independent noise; and **"act perturb"** stands for activity perturbation with independent noise. Note that indepedent weight perturbation is much more costly to compute in

neural networks. As shown in the figure, the empirical variances match very well with our theoretical predictions.

## 11 TRAINING DETAILS

Here we provide more training details.

**MNIST.** We use a batch size of 128, and the SGD optimizer with learning rate 0.01 and momentum 0.9 for a total of 1000 epochs with no data augmentation and a linear learning rate decay schedule.

**CIFAR-10.** For the supervised experiments, we use a batch size of 128 and the SGD optimizer with learning rate 0.01 and momentum 0.9 for a total of 200 epochs with no data augmentation and a linear learning rate decay schedule. For the contrastive M/8 experiments, we use a batch size of 512 and the SGD optimizer with learning rate 1.0 and momentum 0.9 for a total of 1000 epochs with BYOL data augmentation using area crop lower bound to be 0.5 and a cosine decay schedule with a warm-up period of 10 epochs. For the contrastive L/8 experiments, we use a batch size of 2048 and the SGD optimizer with learning rate 4.0 and momentum 0.9 for a total of 1000 epochs with BYOL data augmentation (Grill et al., 2020) using area crop lower bound to be 0.3 and a cosine decay schedule with a warm-up period of 10 epochs.

**ImageNet.** For the supervised experiments, we use a batch size of 256 and the SGD optimizer with learning rate 0.05 and momentum 0.9 for a total of 120 epochs with BYOL data augmentation (Grill et al., 2020) using area crop lower bound to be 0.3 and a cosine learning rate decay schedule with a warm-up period of 10 epochs. For the contrastive experiments, we use a batch size of 2048 and the LARS optimizer with learning rate 0.1 and momentum 0.9 for a total of 800 epochs with BYOL data augmentation (Grill et al., 2020) using area crop lower bound to be 0.08 and a cosine learning rate decay schedule with a warm-up period of 10 epochs.

## 12 FUSED JVP/VJP DETAILS

In Algorithm 1, we provide a JAX code snippet implementing fused operators for the supervised cross entropy loss. "Fused" here means that we package several operations into one function. In the supervised cross entropy loss, we combine average pooling, channel concatenation, a linear classifier layer, and cross entropy all together. Key steps and expected tensor shapes are annotated in the comments. The fused InfoNCE loss implementation will be included in our full code release.

## 13 LOCALMIXER ARCHITECTURE

In Algorithm 2, we provide code in JAX style that implements our proposed LocalMixer architecture.

## 14 ADDITIONAL RESULTS

In this section we provide additional experimental results.

**Normalization scheme.** Table 5 compares different normalization schemes. Layer normalization (LN) is often better than batch normalization (BN) on our mixer architecture. Local LN is better on contrastive learning experiments and achieves lower error rates using forward gradient learning. Although in our main paper, backprop were used in normalization layers, backprop is not necessary for Local LN, $c.f$. "NG" (No Gradient) columns in Table 5.

**Place of normalization.** We investigate the places where we add normalization layers. Traditionally, normalization is added after linear layers. In MLPMixer, LN is added at the beginning of each block. With our forward gradient learning, it is now a question of which location is the optimal design. Adding it after the linear layer has the advantage of shaping the activations to be more well behaved, which can make perturbation learning more effective. Adding it before the linear layer can also help reduce the variance since the inputs always get multiplied with the gradient of the output activity. The results are reported in Table 6. Adding normalization both before and after the linear layer helps forward gradient to achieve lower training errors. While this could result in some overfitting on supervised learning, it is good for contrastive learning which needs more model capacity. This is reasonable as forward gradient introduce a lot of variances, and more normalization layers help achieve better training performance.

---

**Algorithm 1** Naïve and fused local cross entropy, with custom JVP and VJP operators.

```python
# N: batch size; P: num patches; G: num grps; C: num channels; D: channels / grp; K: num cls
# x: encoder features [N,P,G,C/G]
# w: classifier weights [C,K]; b: classifier bias [K]
# labels: class labels [N,K]
import jax
import jax.numpy as jnp
from jax.scipy.special import logsumexp

def naive_avg_group_linear_xent(x, w, b, labels):
    N, P, G, _ = x.shape
    # Average pooling, with stop gradients. [N,P,G,C/G] -> [N,1,G,C/G]
    avg_pool_p = jnp.mean(x, axis=1, keepdims=True)
    x_div_p = x / float(P)
    # [N,P,G,C/G]
    x = x_div_p + jax.lax.stop_gradient(avg_pool_p - x_div_p)
    # Concatenate everything, with stop gradients. [N,P,G,C] -> [N,P,G,G,C/G]
    x = jnp.tile(jnp.reshape(x, [N, P, 1, G, -1]), [1, 1, G, 1, 1])
    mask = jnp.eye(G)[None, None, :, :, None]
    x = mask * x + jax.lax.stop_gradient((1.0 - mask) * x)
    # [N,P,G,G,C/G] -> [N,P,G,C]
    x = jnp.reshape(x, [N, P, G, -1])
    logits = jnp.einsum('npgc,cd->npgd', x, w) + b
    logits = logits - logsumexp(logits, axis=-1, keepdims=True)
    loss = -jnp.sum(logits * labels[:, None, None, :], axis=-1)
    return loss

def fused_avg_group_linear_xent(x, w, b, labels):
    # This is for forward pass. The numerical value of each local loss should be the same.
    # So we compute one and replicate it many times.
    N, P, G, _ = x.shape
    # [N,P,G,C/G] -> [N,G,C/G]
    x_avg = jnp.mean(x, axis=1)
    # [N,G,C/G] -> [N,C]
    x_grp = jnp.reshape(x_avg, [x_avg.shape[0], -1])
    # [N,C] -> [N,K]
    logits = jnp.einsum('nc,ck->nk', x_grp, w) + b
    logits = logits - logsumexp(logits, axis=-1, keepdims=True)
    loss = -jnp.sum(logits * labels, axis=-1)
    # Key step: after computing the loss, replicate it for PxG times. [N] -> [N,P,G]
    return jnp.tile(jnp.reshape(loss, [N, 1, 1]), [1, P, G])

def fused_avg_group_linear_xent_jvp(primals, tangents):
    # This JVP operator performs both regular forward pass and the forward autodiff.
    x, w, b, labels = primals
    dx, dw, db, dlabels = tangents
    N, P, G, D = x.shape
    dx_avg = dx / float(P)
    # Reshape the classifier weights, since only one group passes gradient at a time.
    w_ = jnp.reshape(w, [G, D, -1])
    b = jnp.reshape(b, [-1])
    # Regular forward pass
    # [N,P,G,C/G] -> [N,G,C/G]
    x_avg = jnp.mean(x, axis=1)
    # [N,G,C/G] -> [N,C]
    x_grp = jnp.reshape(x_avg, [x_avg.shape[0], -1])
    # [N,C] -> [N,K]
    logits = jnp.einsum('nd,dk->nk', x_grp, w) + b
    logits = logits - logsumexp(logits, axis=-1, keepdims=True)
    loss = -jnp.sum(logits * labels, axis=-1)
    # We can compute the gradient through cross entropy first.
    dlogits_bwd = jax.nn.softmax(logits, axis=-1) - labels # [N,K]
    # Key step: dloss = dx * w * dloss/dlogit + (x * dw + db) * dloss/dlogit
    # Do the einsum together to avoid replicating outputs.
    dloss = jnp.einsum('npgd,gdk,nk->npg', dx_avg, w_, dlogits_bwd) + jnp.einsum('nk,nk->n',
        (jnp.einsum('nc,ck->nk', x_grp, dw) + db), dlogits_bwd)[:, None, None] # [N,P,G]
    # Return loss and loss gradients [N,P,G].
    return jnp.tile(jnp.reshape(loss, [N, 1, 1]), [1, P, G]), dloss

def fused_avg_group_linear_xent_vjp(res, g):
    # This is a fused backprop (VJP) operator.
    x, w, logits, labels = res
    N, P, G, D = x.shape
    x_avg = jnp.mean(x, axis=1)
    x_grp = jnp.reshape(x_avg, [x_avg.shape[0], -1])
    # Key step: only the first patch/group gradients since everything is the same.
    g_ = g[:, 0:1, 0]
    dlogits = g_ * (jax.nn.softmax(logits, axis=-1) - labels) # [N,K]
    # Remember to multiply gradients by PG times due to weight sharing.
    db = jnp.reshape(jnp.sum(dlogits, axis=[0]), [-1]) * float(P * G)
    dw = jnp.reshape(jnp.einsum('nc,nk->ck', x_grp, dlogits),
        [G * D, -1]) * float(P * G)
    # Key step: use grouped weights to perform backprop.
    dx = jnp.einsum('nd,gcd->ngc', dlogits, jnp.reshape(w, [G, C, -1])) / float(P)
    # Broadcast gradients across patches.
    dx = jnp.tile(dx[:, None, :, :], [1, P, 1, 1])
    return dx, dw, db, None
```

---

**Algorithm 2** A LocalMixer architecture implemented with JAX style code.

```
import jax
import jax.numpy as jnp

def linear(x, w, b):
    """Linear layer."""
    return jnp.einsum('npc,cd->npd', x, w) + b

def group_linear(x, w, b):
    """Linear layer with groups."""
    return jnp.einsum('npgc,gcd->npgd', x, w) + b

def normalize(x, axis=-1, eps=1e-5):
    """Normalization layer."""
    mean = jnp.mean(x, axis=axis, keepdims=True)
    mean_of_squares = jnp.mean(jnp.square(x), axis=axis, keepdims=True)
    var = mean_of_squares - jnp.square(mean)
    inv = jax.lax.rsqrt(var + eps)
    y = (x - mean) * inv
    return y

def block0(x, params):
    """Initial block with only channel mixing."""
    N, P, _ = x.shape
    G = num_groups
    x = normalize(x)
    x = linear(x, params[0][0], params[0][1])
    x = normalize(x)
    x = jax.nn.relu(x)
    x = jnp.reshape(x, [N, P, G, -1])
    x = normalize(x)
    x = group_linear(x, params[1][0], params[1][1])
    x = normalize(x)
    x = jax.nn.relu(x)
    return x

def mlp_block(x, params):
    """Regular MLP block with token & channel mixing."""
    N, P, G, _ = x.shape
    inputs = x
    # Token mixing.
    x = jnp.reshape(x, [N, P, -1])
    x = normalize(x)
    x = jnp.swapaxes(x, 1, 2)
    x = linear(x, params[0][0], params[0][1])
    x = jnp.swapaxes(x, 1, 2)
    x = normalize(x)
    x = jax.nn.relu(x)

    # Channel mixing.
    x = normalize(x)
    x = linear(x, params[1][0], params[1][1])
    x = normalize_layer(x)
    x = jax.nn.relu(x)
    x = jnp.reshape(x, [N, P, G, -1])
    x = normalize(x)
    x = group_linear(x, params[2][0], params[2][1])
    x = normalize(x)
    x = x + inputs
    x = jax.nn.relu(x)
    return x

def local_mixer(x, params):
    """LocalMixer."""
    x = preprocess(x, image_mean, image_std, num_patches)
    pred_local = [] # Local predictions.
    # Build network blocks.
    for blk in range(num_blocks):
        if blk == 0:
            x = block0(x, params[f'block_{blk}'])
        else:
            x = mlp_block(x, params[f'block_{blk}'])

        # Projector connects to local losses.
        x_proj = normalize(x)
        pred_local.append(linear(x_proj, params[f'proj_{blk}'][0], params[f'proj_{blk}'][1]))

        # Disconnect gradients.
        x = jax.lax.stop_gradient(x)
    x = jnp.reshape(x, [x.shape[0], x.shape[1], -1])
    x = jnp.mean(x, axis=1) # [N,C]
    x = normalize(x)
    pred = linear(x, params['classifier'][0], params['classifier'][1])
    return pred, pred_local
```

| | Supervised M/8/16 | | | | Contrastive M/8/16 | | | |
|---|---|---|---|---|---|---|---|---|
| | BP | LG-BP | LG-FG-A | LG-FG-A (NG) | BP | LG-BP | LG-FG-A | LG-FG-A (NG) |
| BN | **30.38 / 0.00** | 33.41 / 5.41 | 32.84 / 23.09 | 33.48 / 20.80 | 27.56 / 24.39 | 30.27 / 28.03 | 35.47 / 32.71 | 37.41 / 31.52 |
| LN | 30.55 / 0.00 | **33.17 / 7.09** | **29.03 / 17.63** | **29.33** / 19.26 | 23.52 / **20.71** | **27.41 / 24.73** | 34.21 / 31.38 | 36.24 / 34.12 |
| Local LN | 32.89 / 0.00 | 33.84 / **0.05** | 30.68 / 19.39 | 30.44 / **17.12** | **23.24** / 21.03 | 28.42 / 25.20 | **32.89 / 31.01** | **32.25 / 30.17** |

Table 5: Comparing different normalization schemes. NG=No normalization gradient. CIFAR-10 test / train error (%)

| LN | Supervised M/8/16 | | | Contrastive M/8/16 | | |
|---|---|---|---|---|---|---|
| | BP | LG-BP | LG-FG-A | BP | LG-BP | LG-FG-A |
| Begin Block | 31.43 / **0.00** | 34.73 / 1.45 | 34.89 / 31.11 | 23.27 / 20.69 | **25.82 / 22.96** | 89.93 / 90.71 |
| Before Linear | 32.50 / **0.00** | 34.15 / **0.05** | 33.88 / 27.94 | 22.62 / **20.38** | NaN / NaN | 35.01 / 32.91 |
| After Linear | **30.38 / 0.00** | 29.83 / 0.44 | **29.35** / 23.19 | 26.50 / 23.98 | 28.97 / 26.67 | 34.10 / 33.18 |
| Before + After Linear | 33.62 / **0.00** | 33.84 / **0.05** | 30.68 / **19.39** | 23.24 / 21.03 | 28.42 / 25.20 | **32.89 / 31.01** |

Table 6: Place of LayerNorm on CIFAR-10, test / train error. (%)

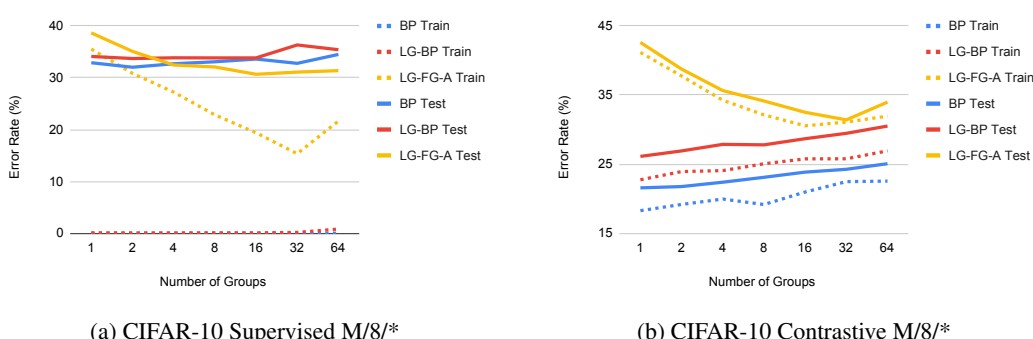

(a) CIFAR-10 Supervised M/8/*          (b) CIFAR-10 Contrastive M/8/*

Figure 9: Effect of groups. For BP algorithms, groups has a minor effect on the final performance, but for local forward gradient, it significantly reduces the variance and achieves lower error rate on both training and test sets.

**Effect of groups.** We provide additional results summarizing the training and test performance of adding more groups in Figure 9. Backprop and local greedy backprop always achieve zero training error with increasing number of groups on CIFAR-10 supervised, but adding groups has a significant benefit lowering training errors for forward gradient. This suggests that the main opponent here is still the gradient estimation variance, and lowering training errors can generally make test errors lower too; on the other hand adding groups have negligible effect on backprop. For contrastive learning, here the task requires higher model capacity, and adding groups effectively reduce the model capacity by introducing sparsity in the weight matrix. As a result, we observe a slight drop of less than 5% performance on both backprop and local greedy backprop. By contrast, forward gradient gains over 10% of performance by adding 16 groups.

