# OpenReview forum: "Scaling Forward Gradient With Local Losses"
_ICLR.cc/2023/Conference — ICLR 2023 poster_

### Official Review · Reviewer_RQfz · 2022-10-24

**Confidence:** 3
**Correctness:** 4
**Technical Novelty And Significance:** 3
**Empirical Novelty And Significance:** 3
**Recommendation:** 8

**Clarity, Quality, Novelty And Reproducibility:**

Clarity:
---
Section 3.2 was easy to follow, perhaps writing that one looks at the directional derivative along a random direction can improve the connection to the previous Section 3.1 which introduces directional gradients. It was a bit unclear why one uses a Gaussian distribution for the perturbation opposed to other distributions? I was also curious if there are connections to mean-field variational inference method which also use weight-perturbation to train NNs (though in a different way).

Section 3.3 was more difficult to understand. In particular, in Equation 3 it is not clear to me how the perturbation in the activations is mapped back to a gradient in the weights.  How is that equation derived?

The proof of the variance of the estimators looks like a quite tedious calculation spanning multiple pages and I could not verify its correctness.

Reproducibility:
---
The result do not appear easily reproducible from the description alone, so it will be important for the code to be released.

**Strength And Weaknesses:**

Strengths:
---
The paper is well-written and was easy to follow, except some small details (see clarity section).

To best of my knowledge, this is the first paper which gives reasonable results on ImageNet-scale problems using a backprop-free algorithm.

The main advantage of the activity perturbation seems to be the reduced variance. Another way of variance reduction is to consider multiple perturbations and averaging them. Perhaps on a smaller dataset, an experiment could be added which shows how the results change under an increasing number of perturbations.

Weaknesses:
---
The method is not evaluated on standard neural architectures (or perhaps did not perform well on them?). But this does not seem like a big weakness to me, since innovations in training algorithms may require also new architectures as current DL models may be "tuned" to work well with backpropagation.

While the proposed backprop-free methods can learn well, it seems still much worse than standard training on standard architectures. I am not too familiar with the proposed architecture, but note that a small MLP trained with back-prop on MNIST reaches ~1.5% test error, which can be further reduced using tricks like drop-out, etc.   I am a bit surprised that the methods have ~2.5% test error on MNIST despite 0% training error. Could this be due to overfitting? Maybe adding a weight-decay regularizer could help.

In Table 3 and 4, training error is also written in "bold face". Ideally, a good training algorithm has similar training and test error indicating that there is no over-fitting to the training set.  This appears to be the case for "FG-W" method, which however does not perform very well.

My suggestion here is simply not bold the training error, as it is not clear to me whether having a small training error is desirable.  If the computational resources are available, it would also be good to re-run the experiments over multiple random seeds and report mean and standard error.

**Summary Of The Paper:**

The paper shows that an algorithm using only forward-mode differentiation can learn well even on ImageNet-scale problems. The key to achieve this result is to use a carefully crafted neural-network architecture with many local loss functions.

**Summary Of The Review:**

While the paper may be lacking in novelty (e.g., activity perturbation has been proposed before), this is one of the first works demonstrating that backprop-free methods can learn well on ImageNet-scale problems. Such results are interesting for the community, and I therefore recommend acceptance.

---

> ### Author Response · Authors · 2022-11-17
> **Response to Reviewer RQfz (part 1)**
>
> Thank you for your positive review comments!
>
> > **Perhaps on a smaller dataset, an experiment could be added which shows how the results change under an increasing number of perturbations.**
>
> Thank you for the suggestion. In Figure 7, we showed more groups (smaller number of perturbation dimensions per loss) leads to better results. In our newly added results in Appendix 10, we also show the numerical variance of the gradient estimator by increasing the input and the hidden dimensions. Bigger variance will directly affect the convergence of stochastic optimization.
>
> > **The method is not evaluated on standard neural architectures (or perhaps did not perform well on them?). But this does not seem like a big weakness to me, since innovations in training algorithms may require also new architectures as current DL models may be "tuned" to work well with backpropagation.**
>
> Just like you pointed out, many current DL models have been tuned to work with backprop for a decade.
>
> We spend a lot of effort to come up with an architecture that could scale well with local perturbation learning. The reason why we didn't use a standard architecture like ResNet is that we needed local losses to spread out evenly across the entire network, and we cannot have too many layers in between two losses. Past works that use ResNet still rely on using backprop within each residual block, whereas in our work we get rid of backprop entirely. We also added channel groups and local normalization to make it work better.
>
> > **I am a bit surprised that the methods have ~2.5% test error on MNIST despite 0% training error. Could this be due to overfitting? Maybe adding a weight-decay regularizer could help.**
>
> For MNIST, there could be some overfitting since we did not apply any data augmentation. We have already added a weight decay regularizer.
>
> > **In Table 3 and 4, training error is also written in "bold face". Ideally, a good training algorithm has similar training and test error indicating that there is no over-fitting to the training set. This appears to be the case for "FG-W" method, which however does not perform very well.**
>
> We will consider your suggestion on the bold face. In this work, the main challenge is to reduce gradient estimation variance. Therefore, overfitting on the training set could be a good sign that the algorithm is able to utilize the gradient information and optimize the objective function.
>
> > **If the computational resources are available, it would also be good to re-run the experiments over multiple random seeds and report mean and standard error.**
>
> Thanks for the suggestions, we will consider them in future versions. For image classification experiments, the standard deviation is typically very small (less than 0.5%) among different runs.
>
> > **It was a bit unclear why one uses a Gaussian distribution for the perturbation opposed to other distributions? I was also curious if there are connections to mean-field variational inference method which also use weight-perturbation to train NNs (though in a different way).**
>
> A Gaussian distribution is necessary to prove the mean and variance properties of our gradient estimator. We need a zero-mean unit-variance independent variable to achieve unbiasedness, and we need higher order statistics of the Gaussian distribution to compute the variance, but other distributions are OK too. We appreciate your comment on the potential relation to mean-field variational inference. We will happily study and include a reference if the reviewer could provide us a pointer to the literature.
>
> > **Section 3.3 was more difficult to understand. In particular, in Equation 3 it is not clear to me how the perturbation in the activations is mapped back to a gradient in the weights. How is that equation derived?**
>
> Equation 3 is directly derived from Equation 2. The gradients to activations multiplying the inputs ($x_i$)  equates to the gradients to weights. The gradients to activations are estimated by adding a perturbation to the activations ($u_j$). Such a perturbation will result in a small scalar change in the loss function, which is represented by the JVP ($\sum_{j’} \nabla z_{j’} u_{j’}$). And we multiply the scalar back to the perturbation to be the gradient estimator.
>
> > **The proof of the variance of the estimators looks like a quite tedious calculation spanning multiple pages and I could not verify its correctness.**
>
> Thank you for raising the concern. In our updated draft, we have added a numerical simulation study in Appendix 10 (page 27). We use a small two layer neural network to test the gradient estimation variance. In Figure 8, the simulation results match exactly with our analytical forms, which suggests that our proof is correct. We will also spend more effort to make the calculation more concise and easier to follow.

---

> > ### Author Response · Authors · 2022-11-17
> > **Response to Reviewer RQfz (part 2)**
> >
> > > **The result do not appear easily reproducible from the description alone, so it will be important for the code to be released.**
> >
> > Thank you for raising the concern. In our updated draft, we included a code release promise in the abstract. We also included Appendix 12, and 13 to show a code snippet for implementing the LocalMixer architecture and the fused loss operators. We hope these efforts address your concern.
> >
> > > **the paper may be lacking in novelty (e.g., activity perturbation has been proposed before)**
> >
> > We updated our draft to reflect that activity perturbation was proposed before only in restricted forms. LeCun et al. (1988) required fan-out to be much smaller than fan-in. Widrow & Lehr (1990) used a one-hidden-layer structure. And Fiete & Seung (2006) used finite differences over time in continuous-time spiking neural networks. None of these papers have combined activity perturbation with forward-mode auto diff. Our other contributions include the variance analysis of the gradient estimators, a continuous-time formulation, a local loss design, and a new local learning architecture. We hope these aspects clarify the novelty of our work.

---

### Official Review · Reviewer_g2TA · 2022-10-24

**Confidence:** 3
**Correctness:** 4
**Technical Novelty And Significance:** 3
**Empirical Novelty And Significance:** 3
**Recommendation:** 8

**Clarity, Quality, Novelty And Reproducibility:**

The paper is technically correct and proposes novel techniques and neural network architecture.

For the most part, the paper is clearly written. Related work is mentioned and properly cited.

Some concerns regarding clarity and reproducibility were raised in the weaknesses section.

**Strength And Weaknesses:**

Strengths:
1. The paper does a good job reviewing past literature and building up to the proposed solution.
2. The solution itself (both LG-FG-A and the arch, LocalMixer), appears to be non-trivial and interesting.
3. For a backprop (BP) free algorithm, the results for large tasks presented in the paper are impressive. The comparison also includes BP-like methods, which helps at illustrating where the gaps in performance are coming from.
4. The paper includes a large variety of ablation tests to showcase the benefit of different components- this is very helpful since the solution includes both a new method and a new neural network architecture, and it would be otherwise difficult to asses the empirical results.
5. The paper closely monitors the biological plausibility of the algorithm, in line with the motivation of the paper of producing a biologically plausible method for training a neural network.


Weaknesses:
1. Despite its key importance to the paper, I found section 4 (Local Losses), to be too lacking in details. As is, the implementation of the local losses is not clear to me.
    * I found the extensive use of the StopGrad operator to be confusing, and I wonder if the section would be clearer, had the paper used partial-derivative notations instead (With the StopGrad included in the implementation details). All in all, I am not sure why replicating the features with different gradient masks for different losses is better than just aggregating the losses (It's makes the algorithm more complex, but appears to be quite similar).
   * The local losses (supervised and contrastive) use a shared linear layer. Is it shared across all local losses in a given layer? How is it trained (for either variation)? Unless I missed something, it is very not intuitive to me that a single shared weight is sufficient to give helpful information for all the different losses in the layer. I would appreciate this point being elaborated upon.
   * The fused implementation seems to be a key to reducing the memory footprint to be scalable, but only the general concept is explained. The paper empirically compares a naive implementation with the fused implementation, without giving sufficient details as to what either of these implementations means. As such, the results in figure 5 are not reproducible.
   * The contrastive method uses two different "views" for the InfoNCE loss, but I did not understand what these views are. Both views share permutation, and neither MNIST/CIFAR include data augmentation, so how are these views different from each other?

2. The submission did not include code. When combined with the previous points, this raises some reproducibility concerns.

3. The trade-offs for using the algorithm, and different variations of it, are not explained. E.g., Figure 5 illustrates the benefit of the "fused" implementation in terms of both memory and runtime, but it is not clear what the cost of adding more groups is when the fused implementation is used.

**Summary Of The Paper:**

The paper proposes a new method for training neural networks, combining forward mode auto-differentiation with directional gradients (forward gradient), weight permutation, and a novel way of using local losses. When combined with a new, complimentary architecture they propose (Local-Mixer), they show that their method is superior to other backprop-free algorithms, on a variety of classification tasks.

**Summary Of The Review:**

I found the paper to be of high quality-- it is well-written, and provides novel methods that will improve the applicability of backdrop-free training.

My main concern is that section 4 was not clear enough, and I am not confident I understood how local losses (a major part of the contribution) were implemented. I intend to increase the score from 6 to 8 once all the details are clarified. (Edit: Updated to 8)

---

> ### Author Response · Authors · 2022-11-17
> **Response to Reviewer g2TA**
>
> Thank you for your detailed and helpful review comments.
>
> > **Despite its key importance to the paper, I found section 4 (Local Losses), to be too lacking in details. As is, the implementation of the local losses is not clear to me.**
>
> Local losses are added to every block, every patch, and every feature group. For the supervised loss, a linear classification layer is added. This linear classification layer is shared across patches and groups, but not across blocks. The reason why it can be shared is that all features are concatenated across groups and averaged across patches. For contrastive loss, a shared linear projector layer is added. There are stop gradients inserted between blocks to make sure the losses are only affecting a local block of parameters.
>
> > **I found the extensive use of the StopGrad operator to be confusing, and I wonder if the section would be clearer, had the paper used partial-derivative notations instead (With the StopGrad included in the implementation details).**
>
> We appreciate your suggestion on using partial derivatives. In Section 3, we’d like the formulation to be general so we used the full gradients, but we will reconsider what is the best way to present in Section 4.
>
> > **All in all, I am not sure why replicating the features with different gradient masks for different losses is better than just aggregating the losses (It's makes the algorithm more complex, but appears to be quite similar).**
>
> The reason why we needed to replicate features with gradient mask is to avoid allowing the perturbation from one feature group to affect the loss of another group (and similarly on the patch level too). The result is indeed quite similar for backprop, but they are needed when using perturbation-based learning.
>
> > **The local losses (supervised and contrastive) use a shared linear layer. Is it shared across all local losses in a given layer? How is it trained (for either variation)?**
>
> Yes, the linear layers are shared across patches and groups, but not shared across blocks. The linear layer is trained with the standard gradient since it is the last layer before the loss function. Therefore the issue of backpropagation is not involved in this layer.
>
> > **The paper empirically compares a naive implementation with the fused implementation, without giving sufficient details as to what either of these implementations means. As such, the results in figure 5 are not reproducible. The submission did not include code. When combined with the previous points, this raises some reproducibility concerns.**
>
> Thank you for the comment. During the rebuttal period, we added a code snippet that implements the fused loss in Appendix 12. We also added JAX code that implements the LocalMixer architecture in Appendix 13.  We promise to release the full code base upon publication, which will include the training loop and other hyperparameters. We added a statement in the abstract that promises code release. We hope these efforts help address your concern.
>
> > **The contrastive method uses two different "views" for the InfoNCE loss, but I did not understand what these views are. Both views share permutation, and neither MNIST/CIFAR include data augmentation, so how are these views different from each other?**
>
> The two views are from standard data augmentation (e.g. random cropping and random color like in SimCLR). We didn't use data augmentation for supervised experiments but for contrastive experiments, we did use standard data augmentation.
>
> > **The trade-offs for using the algorithm, and different variations of it, are not explained. E.g., Figure 5 illustrates the benefit of the "fused" implementation in terms of both memory and runtime, but it is not clear what the cost of adding more groups is when the fused implementation is used.**
>
> There is still a tiny linear cost to the number of groups since we need to replicate the scalar loss across groups in the end. But the coefficient is so minimal that it does not show up in the graph. We have released the actual implementation of the fused operator in Appendix 12 for examination.

---

> > ### Comment · Reviewer_g2TA · 2022-11-17
> > **Re: Response to Reviewer g2TA**
> >
> > Thank you for the response.
> >
> > Please make sure to add important clarifications to the paper.
> >
> > The mathematical difference between accumulating and replicating losses is still unclear to me. Sure, you are preventing perturbations from affecting specific losses, but don't you have other replications of the same losses that are intended to provide gradients for these perturbations? Is this replication still necessary for LG-FG-W (weight permutation) or is this just for activation permutation?
> >
> > That point aside, I found the response to be satisfactory and updated my score to 8.

---

> > > ### Author Response · Authors · 2022-11-18
> > > **Re Reviewer g2TA**
> > >
> > > Thank you very much for raising your score! We will incorporate the above clarifications in the final version.
> > >
> > > > **The mathematical difference between accumulating and replicating losses is still unclear to me. Sure, you are preventing perturbations from affecting specific losses, but don't you have other replications of the same losses that are intended to provide gradients for these perturbations? Is this replication still necessary for LG-FG-W (weight permutation) or is this just for activation permutation?**
> > >
> > > If we understood the question correctly, you are asking whether concatenating (i.e. replicating) features is necessary. Yes, we found it very beneficial on larger experiments, e.g. CIFAR contrastive learning and ImageNet, and we think that only using local features may result in redundancy. Although the losses are "same" in terms of their loss values, they are affected by different local features due to StopGrad, so their forward gradients are different.
> > >
> > > For weight perturbation, having local losses (LG-FG-W) does not seem to bring much benefit, mainly because weights are already shared across different patches and groups. So the local loss design is mainly for activation perturbation, which is what we advocate in the paper.

---

> > > > ### Comment · Reviewer_g2TA · 2022-12-05
> > > > **Re: Replicating Features**
> > > >
> > > > Sorry for the late response.
> > > >
> > > > Going over the code snippets for fused/ naive approach.
> > > >
> > > > 1. The exact implementation of fused is still not clear to me.
> > > > My best understanding so far is that the main function used in forward-mode is:
> > > > "fused_avg_group_linear_xent_jvp",
> > > > with dx being the directional (?) derivative of the Channel-mixing output layer with respect to the activations you perturbed (Assuming we use activation perturbation). I am guessing dw/db need to be the real gradients of w,b with respect to the loss, so some variation of "fused_avg_group_linear_xent_vjp" need to come first to calculate them (With g = 1?). Is that correct?
> > > >
> > > > 2. My main issue with it is that appears as if the fused version is not only better performing than the stopgrad version, but is also more straight forward. Then, why have the naive version be at the center-front of the paper at all? The StopGrad has a  different implications in this case than in the case of BP, which most readers will be familiar with.

---

> > > > > ### Author Response · Authors · 2022-12-05
> > > > > **Re: Re: Replicating Features**
> > > > >
> > > > > 1. `dx` is the directional gradient accumulated from lower layers. In lower layers, `dx` could sometimes be the perturbation of activations, but since we are not perturbing the final layer, `dx` is not the perturbation in this function. `dw` and `db` are also perturbations / directional gradients. Since `x`, `w`, `b` are all input nodes to a linear layer, for completeness, the "tangent" argument should contain three terms. Note that `dw` and `db` are NOT real gradients with respect to the loss. In the experiments, `dw` and `db` are not actually used (so their values are zero). Although the last layer `w` and `b` do use real gradients (from custom VJP), the forward gradient module (custom JVP) does not depend on it. Once you have the custom JVP defined, all you need to do is to call jax.jvp once, and it does one forward pass on both the primal and the tangent.
> > > > >
> > > > > 2. We are not sure why in your comments StopGrad is counted as a separate version as naive. In our Figure 5, "naive" means the naive implementation of the replicated loss, where features are directly replicated, and we don't think having StopGrad here would change the runtime or memory consumption by a lot, although we have already added StopGrad both in "naive" and "fused". In other words, StopGrad and fused implementation are orthogonal to each other.
> > > > >
> > > > > Let us know if our response helps clarify your concerns.

---

> > > > > > ### Comment · Reviewer_g2TA · 2022-12-06
> > > > > > **Re: Replicating Features**
> > > > > >
> > > > > > Thanks for clarifying 1.
> > > > > >
> > > > > > As for 2-
> > > > > > To be clearer, I referred to the StopGrad and Naive version as one. (As implemented in "naive_avg_group_linear_xent")
> > > > > >
> > > > > > I did so following the observation that when using the fused version ("fused_avg_group_linear_xent_jvp"), we seem to get the loss and the directional derivative without applying StopGrad.  Is the StopGrad implemented on dx before the function is called?

---

> > > > > > > ### Author Response · Authors · 2022-12-06
> > > > > > > **Re: Re: Replicating Features**
> > > > > > >
> > > > > > > There is no explicit StopGrad in the fused operator since we never explicitly replicate the features. If we replicate features then it would be much slower. The idea is that the forward pass is entirely shared, and in the gradient pass we used the trick to reshape the weights and multiplies with dx, so that each group gets its own gradient. See the two "key steps" in both jvp and vjp functions. Both of them reshaped the weights so that the group dimension is separate. This call implemented the gradient that is equivalent to having replicated features and having a StopGrad. Thanks for bringing this up, and we will add additional notes to make sure it's clearer.

---

> > > > > > > > ### Comment · Reviewer_g2TA · 2022-12-06
> > > > > > > > **Re: Replicating Features**
> > > > > > > >
> > > > > > > > Thanks. But this brings us back to my original point-- The presentation of the naive approach in the paper is counter-productive. The paper would be clearer if you just present the fused algorithm from the get-go. It is simpler and it is the algorithm that ends up being used anyway.

---

### Official Review · Reviewer_jdBC · 2022-10-25

**Confidence:** 5
**Correctness:** 3
**Technical Novelty And Significance:** 2
**Empirical Novelty And Significance:** 3
**Recommendation:** 8

**Clarity, Quality, Novelty And Reproducibility:**

Clarity: The paper is easy to read.
Quality: The paper studies an interesting problem.
Novelty: The paper builds on Silver et. al 2022, and proposes to augment forward gradient with local losses. The way the local losses are instantiated are very interesting.
Reproducibility: The paper seems easy to reproduce.



**Strength And Weaknesses:**

Strengths:

- The paper is easy to read.
- The paper tackles a very interesting problem.
- The way the local losses are instantiated is interesting, as naive application of local losses does not seem to result in improvement of the results.

Weaknesses:

"Silver et al. (2022) proposed to update the weights based on the directional gradient along a random perturbation direction"

This seems a bit mis-leading. Silver et. al (2022) proposed to update the weights based on the directional gradient along a random as well as along different learned directions (as a result of truncated backprop or as a result of learned critic), which further helps to reduce the variance of the gradient.

**Summary Of The Paper:**

The paper address the scalability issue of forward gradient learning by employing  many different local greedy loss functions (blockwise, patch-wise, and group-wise local losses, and a combination of all three). The paper shows good performance on MNIST and CIFAR-10, and also outperforms other backprop free algorithms on ImageNet.

**Summary Of The Review:**

The paper tackles an interesting problem i.e., thinking about more biologically plausible learning rules as compared to backprop. The reviewer likes the results, as well as clarity of the paper.

---

> ### Author Response · Authors · 2022-11-17
> **Response to Reviewer jdBC**
>
> Thank you for your positive review comments!
>
> > **Silver et. al (2022) proposed to update the weights based on the directional gradient along a random as well as along different learned directions.**
>
> Thank you for pointing this out. We have updated our manuscript to reflect this.

---

### Author Response · Authors · 2022-11-17
**Authors' general response**

We would like to thank all reviewers for their valuable comments. We have addressed most of the suggestions and updated our manuscript, and edits are colored in brown. Changes include:

- For better reproducibility, we have added a statement in the abstract to promise code release upon acceptance.

- Again for better reproducibility, we added code snippets in Appendix 12 and 13 which implement the fused JAX operator and the LocalMixer architecture in JAX.

- To verify the correctness of our proofs, we added a numerical simulation experiment in Appendix 10 using a small neural network. We vary the number of input and hidden dimensions, as well as the batch size, and we measure the variance of the gradient estimation. Numerical simulation results exactly match our analytical forms.

---

### Decision · Program_Chairs · 2023-01-20

**Decision:**

Accept: poster

**Justification For Why Not Higher Score:**

See issues 1,2, and 3 in my meta-review above. If those were indeed addressed in the final version, this paper might be suitable for a spotlight. Since I can't be sure if this would be the case, I recommend acceptance as a poster, but I wouldn't mind too much if the paper would be bumped up to spotlight.

**Justification For Why Not Lower Score:**

It is a clear-cut "accept": all reviewer voted "accept" (8) and I agree.

**Metareview: Summary, Strengths And Weaknesses:**

This works show a few ways to improve the "forward gradient" method, which calculates the gradient in a neural network using forward mode automatic differentiation in random directions: the activity perturbation forward gradient (similarly to Fiete & Seung, 2006), a "local-mixer" architecture specifically designed to help this method, and many local losses.

These improvements significantly reduce the variance in the gradient estimate. Empirically, it matches backprop on MNIST and CIFAR-10 and significantly outperforms other backprop-free algorithms on ImageNet.

All reviewers were rather positive, and the result obtained here are impressive, but it I think has some flaws in its current form, which I hope will be corrected for the final version:
1) Lack of clarity regarding some methods descriptions (as mentioned by reviewers).
2) Lack of reproducibility (though the authors promised to release the code after acceptance).
3) Lack of detailed discussion on the bio-plausibility of the algorithm, compared to previously suggested algorithms. I found this surprising, given that bio-plausibility is the main motivation of this paper.

**Note From Pc:**

if the above contains the word "oral" or "spotlight" please see: "oral" presentation means -> notable-top-5% and "spotlight" means -> notable-top-25%. As stated in our emails, we are disassociating presentation type from AC recommendations